# Grokking and generalization Collapse: Insights from `HTSR` theory

## Abstract

Grokking is a surprising phenomenon in neural network training where test accuracy remains low for an extended period despite near-perfect training accuracy, only to suddenly leap to strong generalization. In this work, we study grokking using a depth-3, width-200 ReLU MLP trained on a subset of MNIST. We investigate it's long-term dynamics under both weight-decay and, critically, no-decay regimes—the latter often characterized by increasing $l^2$ weight norms. Our primary tool is the theory of Heavy-Tailed Self-Regularization (`HTSR`), where we track the heavy-tailed exponent $\alpha$. We find that $\alpha$ reliably predicts both the initial grokking transition and subsequent anti-grokking. We benchmark these insights against four prior approaches: progress measures—Activation Sparsity, Absolute Weight Entropy, and Approximate Local Circuit Complexity —and weight norm ($l^2$) analysis. Our experiments show that while comparative approaches register significant changes, **in this regime of increasing $l^2$ norm, the heavy-tailed exponent $\alpha$ demonstrates a unique correlation with the ensuing large, long-term dip in test accuracy, a signal not reliably captured by most other measures.**

Extending our zero weight decay experiment significantly beyond typical timescales ($10^5$ to approximately $10^7$ optimization steps), **we reveal a late-stage catastrophic generalization collapse ("anti-grokking"), characterized by a dramatic drop in test accuracy (over 25 percentage points) while training accuracy remains perfect**; notably, the heavy-tail metric $\alpha$ uniquely provides an early warning of this impending collapse. Our results underscore the utility of Heavy-Tailed Self-Regularization theory for tracking generalization dynamics, even in the challenging regimes without explicit weight decay regularization.

## 1 Introduction

Grokking is an intriguing phenomenon where a neural network achieves near-perfect training accuracy quickly, yet the test accuracy lags significantly, often near chance level, before abruptly surging towards high generalization [15]. Figure 1 illustrates this for a depth-3, width-200 ReLU MLP trained on a subset of MNIST.

To dissect this phenomenon and uncover deeper dynamics, our primary analytical lens is the recently developed theory of Heavy-Tailed Self-Regularization (`HTSR`), following Martin et.al.[10]. The `HTSR` theory examines the empirical spectral density (ESD) of individual layer weight matrices ($\mathbf{W}$), quantified by the heavy-tailed power law (PL) exponent $\alpha$. We find $\alpha$ provides a sensitive measure of correlation structure within layers, tracking the transition into the grokking phase, and crucially, predicting a subsequent decrease in generalization.

For comparative context, we also investigate several other methodologies:

1. **Weight Norm Analysis:** Motivated by studies like Liu et al. [6], we examine the $l^2$ norm of the weights. We observe that grokking occurs even without weight decay (leading to an increasing norm), suggesting weight norm alone is not a complete explanation, confirming the weight-norm related findings by Golechha [2].

2. **Progress Measures:** We utilize metrics proposed by Golechha [2].—Activation Sparsity, Absolute Weight Entropy, and Approximate Local Circuit Complexity—which capture broader structural and functional changes in the network during training.

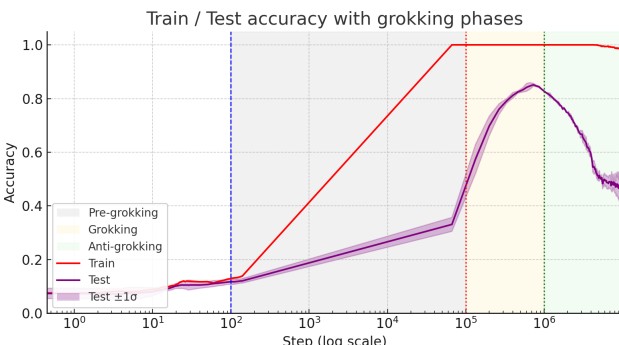

Figure 1: **The three phases of grokking.** Training curves for a depth-3, width-200 MLP on MNIST. The initial **pre-grokking** phase (grey): training accuracy (red line) surges at $10^2$ steps, saturating between $10^4 - 10^5$ steps, while test accuracy (purple line) remains low; the **grokking** phase (yellow): with test accuracy rapidly increasing after $\sim 10^5$ steps, and reaching a maximum at $10^6$ steps; and the newly revealed late-stage **anti-grokking** phase (green): test accuracy collapses (to $0.5$).

**Our Contributions:** Our work makes several related contributions that helps explain the underlying mechanisms associated with the grokking phenomena:

1. By extending training significantly (up to $10^7$ steps) under zero weight decay ($WD = 0$), we identify and characterize **late-stage generalization collapse**: a substantial drop in test accuracy long after initial grokking, despite perfect training accuracy and a continually increasing $l^2$ weight norm. We call this **anti-grokking.**

2. We show that the `HTSR` layer quality metric $\alpha$ (the heavy-tailed power-law (PL) exponent), effectively tracks the grokking transition under both the traditional setting of weight decay ($WD > 0$) and zero weight decay $WD = 0$, outperforming the $l^2$ weight norm and the other progress metrics. Only the `HTSR` $\alpha$ can distinguish between all 3 phases of grokking.

3. We identify the mechanism of the pre-grokking phase, where the training accuracy is perfect but the model does not generalize. This phase occurs because only a subset of the model layers are well trained (i.e. $\alpha \leq 4$), whereas at least one layer is underfit (i.e. $\alpha \geq 5$). Moreover, the layers can show great variability between training runs, indicating their instabili. Importantly, the layer $\alpha$'s here are distinct from those in the anti-grokking phases, despite both phases having perfect training accuracy and low test accuracy. .

4. We demonstrate that when the `HTSR` PL exponent $\alpha < 2$, this identifies the collapse. Also, in this phase, we observe the presence of anomalous rank-one (or greater) perturbations in one or more underlying layer weight matrices $\mathbf{W}$. We call these **correlation traps** and identify them by randomizing $\mathbf{W}$ elementwise, forming $\mathbf{W}^{rand}$, and looking for unusually large eigenvalues, $\lambda_{trap} \gg \lambda^+$ (where $\lambda^+$ is the right-most edge of the associated Marchenko-Pastur (MP) distribution [9]).

## 2   Related Work

Grokking [15], the delayed emergence of generalization well after training accuracy saturation, has prompted significant research into its underlying mechanisms. Initial studies often explored grokking in algorithmic tasks [12, 13, 16], frequently linking the phenomenon to the presence of weight decay (WD) which favors simpler, lower-norm solutions [6]. Other approaches include

mechanistic interpretability [13] and analyses identifying competing memorization and generalization circuits [16, 12].

Varma et al. [16] defined 'ungrokking' as generalization loss when retraining a grokked network on a *smaller* dataset ($D < D_{crit}$), attributing it to shifting circuit efficiencies under WD. In contrast, we observe **late-stage generalization collapse** ('anti-grokking') occurring on the *original* dataset after prolonged training (~$10^7$ steps) *without* WD (WD=0). This distinct phenomenon is not predicted by [16] as it falls outside of the crucial weight decay assumption on which it relies.

Grokking studies now include real-world tasks [2, 4]. Golechha et al. [2] introduced progress measures (e.g., Activation Sparsity, Absolute Weight Entropy) and notably observed grokking without WD, resulting in increasing $l^2$ norms, similar to our setup. We use their metrics for comparison but extend training drastically (up to $10^7$ steps), revealing the subsequent 'anti-grokking' collapse—a phenomenon not reported in their work despite the similar WD=0 regime.

We employ the theory of Heavy-Tailed Self-Regularization (`HTSR`) [10, 11], tracking the spectral exponent $\alpha$. We find $\alpha$ predicts both the initial grokking and, uniquely, the subsequent dip and eventual 'anti-grokking' collapse under WD=0. Our contribution lies in identifying and characterizing this anti-grokking phenomenon using $\alpha$ for long-term generalization stability, extending prior work that either required WD or did not explore sufficiently long training horizons.

## 3    Measures and Metrics

### 3.1    Heavy–Tailed Self-Regularization (`HTSR`)

**From weights to spectra.**    For each layer weight matrix $\mathbf{W} \in \mathbb{R}^{N \times M}$, we build the un-centred *correlation* (Gram) matrix

$$\mathbf{X} \;=\; \frac{1}{N}\,\mathbf{W}^{\top}\mathbf{W} \;\in\; \mathbb{R}^{M \times M}. \tag{1}$$

Let $\{\lambda_i\}_{i=1}^{M}$ be the eigenvalues of $\mathbf{X}$. Their empirical spectral density (ESD) is the discrete measure

$$\rho_{emp}(\lambda) \;=\; \frac{1}{M} \sum_{i=1}^{M} \delta\big(\lambda - \lambda_i\big). \tag{2}$$

**Gaussian baseline.**    If the entries of $\mathbf{W}$ are i.i.d. $\mathcal{N}(0, \sigma^2)$, then, in the limit $N \to \infty, M \to \infty$ with aspect ratio $Q = N/M \geq 1$ fixed, $\rho_{emp}(\lambda)$ converges to the Marchenko–Pastur (MP) density [7]

$$\rho_{\mathrm{MP}}(\lambda) = \begin{cases} \dfrac{Q}{2\pi\sigma^2} \dfrac{\sqrt{(\lambda^+ - \lambda)(\lambda - \lambda^-)}}{\lambda}, & \lambda \in [\lambda^-, \lambda^+], \\ 0, & \text{otherwise}, \end{cases} \quad \lambda^{\pm} = \sigma^2\big(1 \pm Q^{-1/2}\big)^2 \tag{3}$$

This provides a principled "null model" against which real, trained weights can be compared. In a well-trained model, the eigenvalues of any layer $\mathbf{W}$ will rarely conform closely to an MP distribution and will almost always have a significant number of large eigenvalues extending beyond any recognizable bulk MP region ($\lambda \gg \lambda^+$) if not being fully heavy-tailed power-law. If, however, we randomize $\mathbf{W}$ elementwise,

$$\mathbf{W} \to \mathbf{W}^{rand} \tag{4}$$

then the elements of $\mathbf{W}^{rand}$ will be i.i.d. by construction, and we expect that the ESD of $\mathbf{W}^{rand}$ can be very well fit to an MP distribution. This is shown below, on in Figure 2 (Right).

**Heavy–Tailed Self-Regularization (`HTSR`) Theory**    Prior work[10, 11, 9] shows that the ESD of real-world DNN layers with learned correlations almost never sits entirely within the Marchenko–Pastur bulk predicted for i.i.d. Gaussian weights; instead, the right edge flares into a power law (PL) tail . Formally,

$$\rho_{emp}(\lambda) \;\sim\; \lambda^{-\alpha}, \qquad \lambda_{\min} < \lambda < \lambda_{\max}, \tag{5}$$

with the exponent $\alpha$ quantifying the strength of the correlations. According to the `HTSR` framework [11], different ranges of $\alpha$ correspond to the different phases of training and different levels of convergence for each layer:

- $\alpha \gtrsim 5 - 6$: **Random-like or Bulk-plus-Spikes** — the spectrum is close to the Gaussian baseline; little task structure is present.
- $2 \lesssim \alpha \lesssim 5 - 6$: **Weak (WHT) to Moderate Heavy (Fat) Tailed (MHT)** — correlations build up; layers are well-conditioned and typically generalise better.
- $\alpha = 2$ **Ideal value:** Corresponds to fully optimized layers in models. Associated with layers in models that generalize best.
- $\alpha < 2$: **Very-Heavy-Tailed (VHT)** — extremely heavy tails indicate potentially over-fitting to the training data and often precede and/or are associated with decreases in the generalization / test accuracy.

Note that the lower bound of $\alpha = 2$ on the Fat-Tailed phase is a hard cutoff, whereas the upper bound $\alpha \gtrsim 5 - 6$ is somewhat looser because it can depend on the aspect ratio $Q$. See Martin et. al. [10, 9] for more details.

**Estimating $\alpha$.** Following [11], we fit the tail of $\rho_{emp}$ to a PL 5 through the maximum likeli-hood estimator (MLE) [1]. The start of the Pl tail, $\lambda_{\min}$, is chosen automatically to minimize the Kolmogorov-Smirnov distance between the empirical and fitted distributions. All calculations are performed with `WeightWatcher` v0.7.5.5 [8], which automates

- SVD extraction of singular values $\sigma_i$ ($\lambda_i = \sigma_i^2$),
- PL fits and goodness-of-fit KS tests (including selection of $\lambda_{\min}$ and $\lambda_{\max}$)
- Detection of correlation traps (optional)

Figure 2 (Left) shows an example of a PL fit on a log-log scale for a representative layer after training. The plot displays the ESD for a typical NN layer (a histogram or kernel density estimate of eigenvalues), the automatically chosen $\lambda_{\min}$ (xmin, vertical line, red), the $\lambda_{\max}$ (xmax, vertical line, orange), and the best fit for the PL tail (dashed line, red).

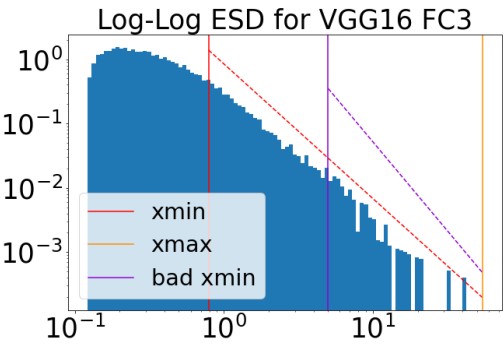 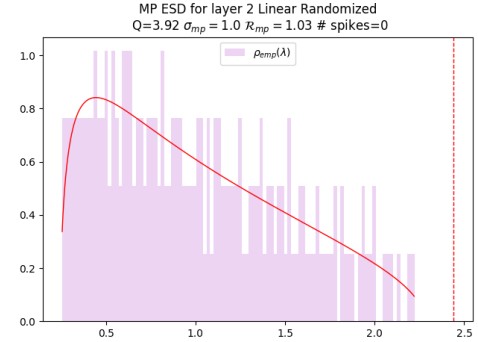

Figure 2: **Left**: Example of the ESD derived from a well-correlated $\mathbf{W}$ (blue) and the Power-Law fit to the tail (red), on a **Log-Log** plot. **Right**: Example of the ESD of $\mathbf{W}^{rand}$ (light purple) and the MP fit (red), on a **Log-Linear** plot.

Note that the PL fit is very sensitive to the choice of $\lambda_{\min}$, and a poor choice will result in a poorly estimated $\alpha$. If $\lambda_{\min}$ is too large (bad xmin, vertical line, purple), then the PL tail is too small and results in a larger $\alpha$. The selection of $\lambda_{\min}$ is very important in the calculation of the tail alpha ($\alpha$) and is fully automated using the open-source `WeightWatcher` tool.[8]

**Significance for Grokking/Anti-Grokking.** Across all experiments, the trajectory $\alpha(t)$ proves to be a highly sensitive indicator of the network's generalization state: large drops toward $\alpha \approx 2$ coincide with the onset of grokking, while a further fall below $\alpha < 2$ foreshadows (and then characterizes) the eventual "anti-grokking" collapse .

## 3.2 Correlation Traps

To better understand the origin of anti-grokking (generalization collapse), it is instructive to look for evidence of potential overfitting in the layer weight matrices $\mathbf{W}$, which appear as what we

call **Correlation Traps** [9]. Recall that for a well-trained model, we expect the ESDs of $\mathbf{W}^{rand}$ to be well-fit by an MP distribution; here we argue that deviations from this are significant and informative. To identify these deviations, we compare the randomized layer ESDs against the MP distribution at the different stages of training to assess deviations from randomness. We identify these deviations as anomalously large eigenvalues in the underlying $\mathbf{W}^{rand}$. We call such large eigenvalues correlation traps, $\lambda_{trap}$, when they are significantly larger than the bulk edge $\lambda_{rand}^+$ of the best fit MP distribution.

$$\lambda_{trap} \gg \lambda_{rand}^+ \tag{6}$$

See the Appendix D for additional statistical validation of the presence of such traps, as well as the Supplementary Information. Also, see [9] for more details.

The `WeightWatcher` tool [8] detects correlation traps automatically; it randomizes $\mathbf{W}$, then performs automated MP fits by estimating the variance $\sigma_{MP}^2$ of the underlying randomized matrix $\mathbf{W}^{rand}$, finding the fit that best describes the bulk of its ESD of $\mathbf{W}^{rand}$. It then finds all eigenvalues $\lambda_{trap}$ that are significantly larger (i.e. beyond the Tracy-Widom fluctuations) of the MP bulk edge $\lambda_{rand}^+$ of the ESD of $\mathbf{W}^{rand}$. Figure 3 depicts two layers from the models studied here with correlation traps.

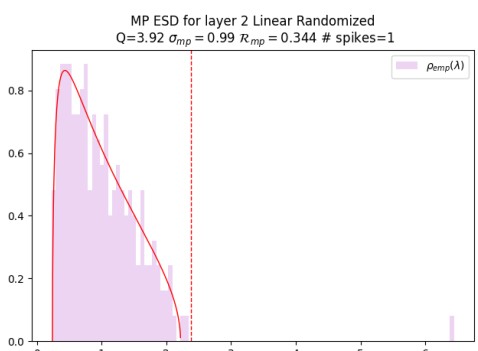 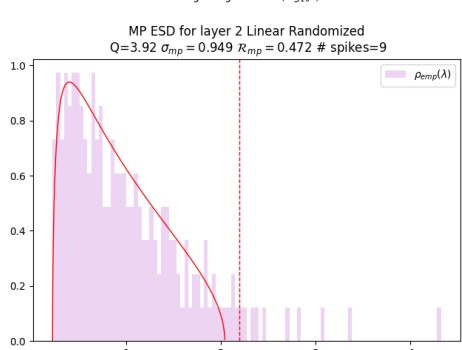

Figure 3: **Examples of Correlation Traps**. ESDs of ($\mathbf{W}^{rand}$) (light purple) of Layer 2 for the randomized weight matrix $\mathbf{W}^{rand}$ for different models, compared to an MP fit (red). Correlation traps $\lambda_{trap}$ are depicted a small spikes to the right of the MP fit. (x-axis is log scale) **Left: Right Before Collapse** (i.e. at more than $\sim 10^6$ steps) ($\sigma_{mp} \approx 0.9879$). The KS test (P-value $\approx 4 \times 10^{-13}$) indicates a strong deviation from the MP model. A single, prominent correlation trap appears at $\lambda_{trap} \approx 10^{6.5}$. **Right: Final Generalization Collapse**. The KS test (P-value $\approx 1.877 \times 10^{-5}$) indicates a strong deviation from the MP model. Multiple correlation traps are observed, $\lambda_{trap} \in [10^{2.x}, 10^{6.5}]$.

For additional statistical validation, here, we also use the Kolmogorov-Smirnov (KS) test to quantify the dissimilarity between the ESD of $\mathbf{W}^{rand}$ and its best MP fit. A large difference, combined with a visual inspection of the data, indicates the presence of one or more correlation traps ($\lambda_{trap}$).

## 3.3 Other Benchmarked Metrics

We benchmarked our `HTSR`-based findings against $l^2$ weight norm analysis [6] and several progress measures proposed by Golechha [2], these include Activation Sparsity ($A_s$), Absolute Weight Entropy ($H_{abs}(W)$), and Approximate Local Circuit Complexity ($\Lambda_{LC}$). Detailed definitions of these measures are provided in Appendix B.

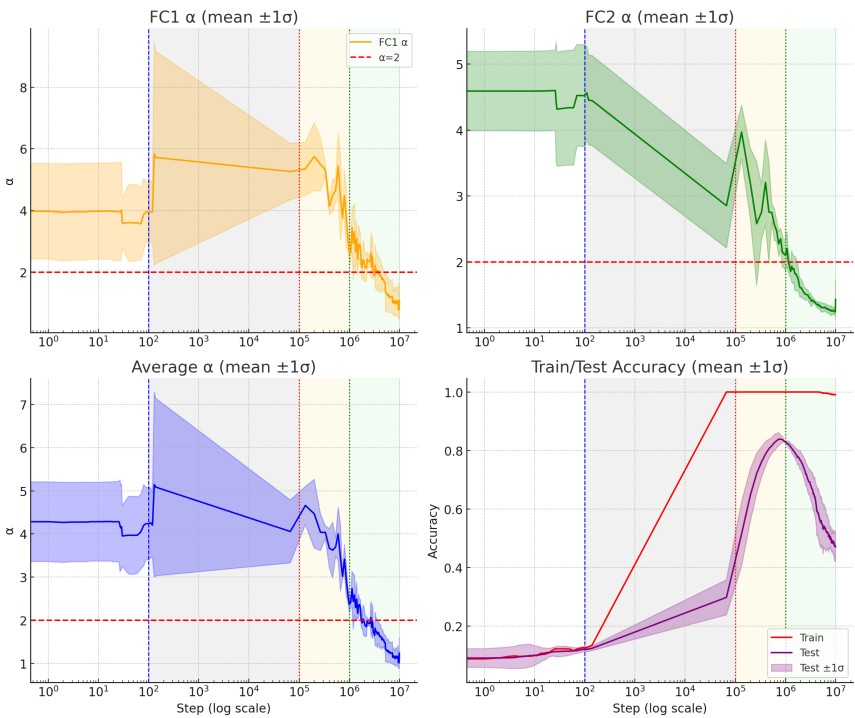

Figure 4: `HTSR` results vs. optimization steps. Top: Average $\alpha$ across layers. Middle: $\alpha$ for the first fully connected layer (FC1). Bottom: $\alpha$ for the second fully connected layer (FC2). Note the significant dip below the critical threshold $\alpha = 2$, especially in FC2, coinciding with the "anti-grokking" performance drop seen in Fig. 1 after 1M steps.

## 4  Results and Analysis

### 4.1  Layer Metrics for Tracking Grokking

**HTSR layer quality metric $\alpha$:**  Our primary metric, the `HTSR` layer quality metric $\alpha$, reveals critical dynamics missed by other measures. Figure 4 shows the evolution of $\alpha$ averaged across layers (top) and for individual fully connected layers (middle, bottom).

Table 1: **Layer-wise and average `HTSR` $\alpha$ exponents.**  At the right edge of each grokking phase: Pre-grokking $\sim 10^5$ steps, Grokking $10^6$ steps, and Anti-grokking $10^7$ steps, For the zero-weight-decay ($WD = 0$) experiment; values are taken from Fig. 4. Various seeds are used and variability in initialization, optimizer trajectory may occur.

| Layer, Metric | Pre-grokking | Grokking (Max Test Acc.) | Anti-grokking (Collapse) |
|---|---|---|---|
| FC1 $\alpha$ | $4.0 \pm 1.3$ | $3.2 \pm 0.6$ | $1.0 \pm 0.40$ |
| FC2 $\alpha$ | $4.6 \pm 0.5$ | $2.4 \pm 0.1$ | $1.4 \pm 0.24$ |
| average $\alpha$ | $4.3 \pm 0.70$ | $2.8 \pm 0.30$ | $1.2 \pm 0.23$ |

Initially, $\alpha$ is high, reflecting random-like weights. As training progresses and the network begins to fit the training data, $\alpha$ decreases. The sharp drop towards the optimal (fat-tailed) regime ($2 \lesssim \alpha \lesssim 5-6$) coincides with the rapid improvement in test accuracy characteristic of grokking (around $10^4$-$10^5$ steps in Figure 1). Crucially, as training continues into the millions of steps, $\alpha$ consistently dips below 2, entering the Very Heavy-Tailed (VHT) regime. This occurs notably in the second fully connected layer (FC2, bottom panel). This drop below $\alpha = 2$, indicating potential layer non-optimality and overly strong correlations, directly precedes and coincides with the significant drop in test accuracy—the "anti-grokking" phase—observed after $10^6$ steps in Figure 1.

Together, these observations highlight the unique sensitivity of the `HTSR` $\alpha$ metric. This metric not only identifies the grokking transition but also provides an early warning for the subsequent instability and the novel "anti-grokking" phenomenon, highlighting potentially pathological correlation structures forming deep into training. The layer-wise analysis (Figure 4) further suggests that this instability might originate in specific layers (i.e. FC2 here) becoming over correlated ($\alpha < 2$).

**Comparative metrics:** In contrast, the comparative metrics capture the initial training and grokking phases but fail to predict the late-stage generalization collapse. Figure 5 displays the Activation Sparsity, Absolute Weight Entropy, and Approximate Local Circuit Complexity. While these metrics show clear trends during the initial learning and grokking phases (e.g., changes in sparsity and complexity), their trajectories become relatively stable or lack distinct features corresponding to the dramatic performance drop seen during "anti-grokking". For example, circuit complexity remains relatively flat in the late stages up until some noise at the end, offering no warning of the impending collapse. Though Activation Sparsity shows an inflection around peak test accuracy and does detect grokking, it generally continues its upward trend through the late-stage collapse.

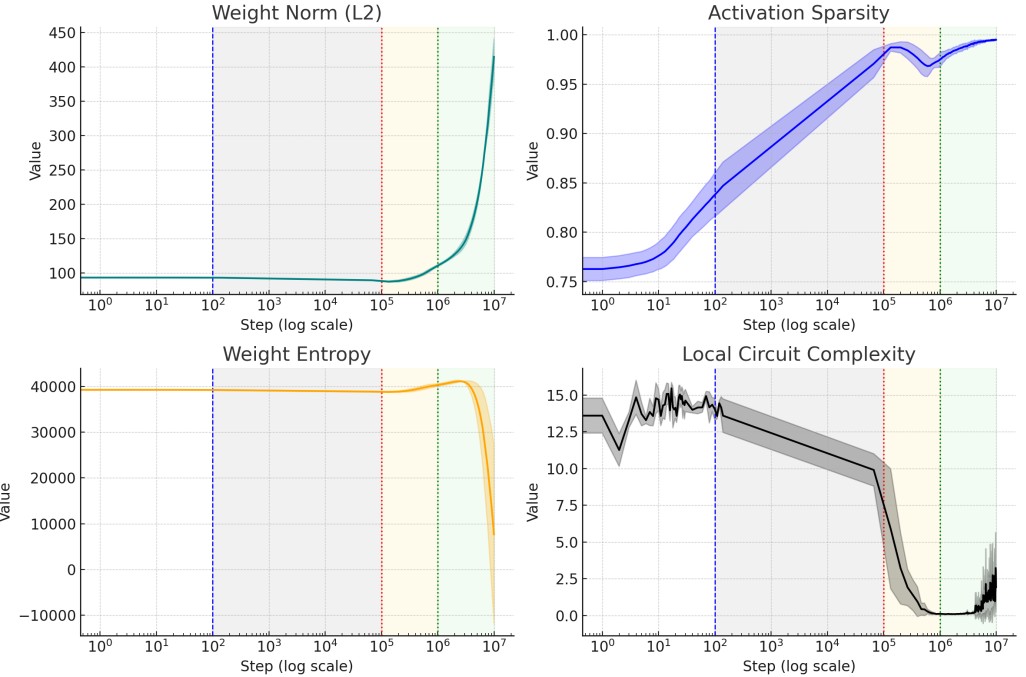

Figure 5: Alternative progress measures (Golechha [2]) vs. optimization steps. Top: Activation Sparsity. Middle: Absolute Weight Entropy. Bottom: Approximate Local Circuit Complexity. While these metrics show changes during the initial training and grokking phases (Activation Sparsity for example), they do not exhibit clear signals predicting the magnitude of the late-stage "anti-grokking" performance dip observed after $10^6$ steps.

In our primary WD=0 experiments, $A_s$ generally increases throughout training (Figure 5), seemingly tracking the pre-grokking and grokking phases, however, it fails the negative control in the anti-grokking phase because it continues to increase in the same way as in pre-grokking. Prior studies have linked activation sparsity to generalization [5, 12, 14] and reported specific dynamics such as plateauing before grokking [2] or an increase preceding a rise in test loss [3]. Specifically, we observe a subtle inflection or dip in $A_s$ coinciding with the point of maximum test accuracy before a slight increase. While this feature appears to mark a shift around peak test accuracy, its specific predictive utility for subsequent generalization dynamics is questionable. In other words, without knowing the proper sparsity cutoff, it is impossible to determine if increasing $A_s$ corresponds to pre-grokking or anti-grokking. In contrast, because the `HTSR` $\alpha = 2$ is a theoretically established universal cutoff, one can distinguish between the two phases correctly.

Additionally, in our WD=0.01 control experiment, as detailed in Appendix C, a similar inflection in $A_s$ occurs where test accuracy, after a slight initial decrease from its peak, subsequently plateaus rather than undergoing a catastrophic collapse as seen in the WD=0 case. Therefore, observing this dip in $A_s$ alone does not allow one to distinguish whether test accuracy will catastrophically decline or stabilize, suggesting it primarily indicates that some form of transitional change has occurred around the point of maximum generalization, rather than predicting the specific nature of the subsequent trajectory. Our findings indicate limitations in the other two comparitive metrics for tracking the anti-grokking phase. Absolute Weight Entropy ($H_{abs}(\mathbf{W})$), despite its suggested link to generalization [2], also decreases sharply during the collapse, thus not reliably distinguishing this anti-grokking phase. Similarly, $\Lambda_{LC}$ [2] remains low throughout the collapse, failing to reflect the performance degradation. We also confirm, consistent with [2], that grokking occurs robustly even with increasing weight norms and no weight decay.

## 4.2 Correlation Traps and Anti-Grokking

To better understand the origin of anti-grokking (generalization collapse), it is instructive to look for evidence of potential overfitting in the layer weight matrices $\mathbf{W}$, in the form correlation traps. As described in Section 3.1, we analyze the eigenvalues $\{\lambda_i\}$ of the randomized weight matrices $\mathbf{W}^{rand}$ derived from each layer's weight matrix $\mathbf{W}$ for layers FC1 and FC2.

Table 2: **Average number of detected correlation traps** in layers FC1 and FC2 at the right edge of of the three grokking phases: Pre-grokking $\sim 10^5$ steps, Grokking $10^6$ steps, and Anti-grokking $10^7$ steps. Results shown for both experiments, with ($WD > 0$) and without $WD = 0$ weight decay.

| Model, Layer | Pre-grokking | Grokking (Max Test Acc.) | Anti-grokking (Collapse) |
|---|---|---|---|
| $WD = 0$, FC1 | 0 | 0 | $6.33 \pm 5.44$ |
| $WD = 0$, FC2 | 0 | 0 | $1.00 \pm 0.00$ |
| $WD > 0$, FC1 | 0 | 0 | $2.00 \pm 0.00$ |
| $WD > 0$, FC2 | 0 | 0 | $1.00 \pm 0.00$ |

As show in Table 2, for both layers, FC1 and FC2, and for both experiments, with and without weight decay, neither layer shows evidence of correlation traps until the anti-grokking phase. The presence of such traps corresponds to `HTSR` $\alpha < 2$ for these layers, as predicted by previous work[9]. Further statistical analysis for the FC2 layers is provided in Appendix D. The presence of correlation traps, combined with $\alpha < 2$, is a definitive signal indicating the model is in the anti-grokking phase.

## 5 Conclusion

This study investigated the well-known grokking phenomena in neural networks (NN) under the lens of the recently developed theory of Heavy-Tailed Self Regularization (`HTSR`) [10]. Previous work has attempted to explain grokking (using the $l^2$ norm), but only succeeds in the presence of weight decay (WD), and has been unable to explain grokking without weight decay[6, 2]. For this reason, we have studied the long-term generalization dynamics of the grokking phenomena both with weight decay ($WD > 0$) and without ($WD = 0$). We compare the application of the `HTSR` theory to the $l^2$ norm and several previous proposed metrics.[6] Our primary finding is that the `HTSR` layer quality metric $\alpha$ can effectively track grokking both with and without weight decay. In particular, the `HTSR` $\alpha$ tracks the initial grokking transition and subsequent performance dips in both case ($WD = 0$ , $WD > 0$) and, in doing so, offers new insights into the grokking phenomena.

Moreover, and critically, in the $WD = 0$ setting, the `HTSR` $\alpha$ also provides an early indication of a novel **late-stage generalization collapse**, called **anti-grokking**. This collapse is characterized by a significant drop in test accuracy despite sustained perfect training accuracy (and a large $l^2$ norm), and is observed after extensive training (up to $10^7$ steps).

We also examined several other grokking progress measures, in addition to the $l^2$ norm [6], including Activation Sparsity $A_s$, Absolute Weight Entropy $H_{abs}(W)$, and Approximate Local Circuit Complexity $\Lambda_{LC}$ [2]. Although $A_s$ and $\Lambda_{LC}$ captured initial training and grokking phases, and do

change at the anti-grokking transition, they failed to unambiguously predict the appearance and/or presence of anti-grokking.

In examining the `HTSR` results on all 3 phases of grokking, we propose a new explanation of the grokking phenomena. During the first phase, pre-grokking, where only training accuracy saturates, only a subset of the individual layers will converge, and only far enough (i.e, $\alpha \approx 4$) to describe the training data, while other layers will appear almost random (i.e, $\alpha \approx 5$). Importantly, some layers will be more important for generalization than others, and these will not have converged very well at all. During the grokking phase, when the test accuracy is maximal, all important layers will converge extremely well, with $\alpha$ metrics approach the optimal value with $\alpha \approx 2.0$–exactly as predicted by the `HTSR` theory. In the third anti-grokking phase, where the test accuracy drops substantially, one or more layer will overfit the data in some yet undetermined way). They will have $\alpha < 2$, and may exhibit correlation traps (and/or even rank collapse). (Note these results are also supported by recent theoretical developments in `HTSR` (and SETOL) theory[9].)

In particular, we consider the implications of observing numerous correlation traps in the anti-grokking phase. The 'traps' are anomalous rank-one (or greater) perturbations in the weight matrix $\mathbf{W}$, causing a large mean-shift in underlying distribution of elements: $\mathbb{E}[W_{ij}] \to$ *large* and, 'pushing' the ESD into the VHT phase where $\alpha < 2$. The large shift in $\mathbb{E}[W_{ij}] \to$ *large* indicates that the distribution of weights is *atypical*. That is, different random samples of the weights could have very different means. And as with any statistical estimator, an atypical distribution will not generalize well. (Similar results have been seen in training a similar model with very large learning rates[9].) Consequently, it is hypothesized that layers with large numbers of correlation traps are overfit to the training data (in some unspecified way), and hurt the overall model test accuracy.

These results underscore the utility of `HTSR` for monitoring and understanding long-term generalization stability across different regularization schemes, with a particular strength in identifying potential catastrophic collapse. The observed layer-specific changes in $\alpha$ during the $WD = 0$ collapse suggest that potential over-fitting may develop deep into training. While our current findings are based on a specific MLP architecture and MNIST subset, further research should validate these observations across diverse datasets, architectures, hyperparameter configurations, and optimizers. Promising future work includes developing $\alpha$-guided adaptive training strategies. Additionally, designing differentiable regularizers or loss terms based on $\alpha$ could potentially enable faster and more stable generalization, for instance, by encouraging convergence towards $\alpha \approx 2$.

## 6 Limitations

Our study, while providing insights into generalization dynamics via Heavy-Tailed Self-Regularization (`HTSR`), has limitations that define important avenues for future research. The empirical findings are primarily derived from a specific three-layer MLP architecture trained on an MNIST subset. Consequently, the generalizability of the observed $\alpha$ trajectories and their specific predictive power for phenomena like grokking and late-stage generalization collapse warrants further validation across a wider range of model architectures (e.g., CNNs, Transformers), datasets, tasks, and diverse training configurations, including different optimizers and hyperparameter settings.

Furthermore, `HTSR` is an empirically-grounded, phenomenological framework, supported theoretically with a novel application of Random Matrix Theory (RMT). While its correlations between the heavy-tailed PL exponent $\alpha$ and network generalization states are compelling, the interpretation requires careful consideration of context. For instance, while well-generalized models often exhibit $\alpha$ values within the range (e.g., $2 \leq \alpha \leq 6$), and $\alpha \approx 2$ is frequently associated with optimal performance or critical transitions, this is not a strictly bidirectional implication. It is conceivable that layers or models might exhibit $\alpha$ values near or even below 2 (typically indicating over-correlation) yet display suboptimal generalization. Other very-well trained models may have layers fairly large alphas. This is not yet fully understood. This highlights that while $\alpha$ provides strong correlational insights into learning phases and stability, the precise mapping of specific $\alpha$ values to absolute performance levels can be context-dependent and is an area for ongoing refinement of the theory (see [9]). Our work contributes observations within specific phenomena, acknowledging that the broader applicability and predictive nuances of the `HTSR` theory will benefit from continued exploration.

These limitations underscore the importance of ongoing empirical and theoretical work to further refine, validate, and extend the understanding of `HTSR` theory in deep learning.

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

# Appendices

## A  Experimental Setup

We train a Multi-Layer Perceptron (MLP) on a subset of the MNIST dataset using the hyperparameters detailed in Table 3. The training subset is constructed by randomly selecting 100 samples from each of the 10 MNIST classes, ensuring a balanced dataset of 1,000 unique training points. This was run on an Nvidia Quadro P2000 and took approximately 11 hours. A considerable part of the time is due to the speed of saving the measures.

Table 3: Experimental hyperparameters used in the study (details in Appendix A).

| Parameter | Value |
|---|---|
| Network Architecture | Fully Connected MLP |
| Depth | 3 Linear layers (Input → Hidden1 → Hidden2 → Output) |
| Width | 200 hidden units per hidden layer |
| Activation Function | ReLU (Rectified Linear Unit) |
| Input Layer Size | 784 (Flattened MNIST image $28 \times 28$) |
| Output Layer Size | 10 (MNIST digits 0-9) |
| Weight Initialization | Default PyTorch (Kaiming Uniform for weights), parameters scaled by 8.0 |
| Bias Initialization | Default PyTorch (Uniform), then scaled by 8.0 |
| Dataset | MNIST |
| Training Points | 1,000 (100 per class, stratified random sampling) |
| Test Points | Standard MNIST test set (10,000 samples) |
| Batch Size | 200 |
| Loss Function | Mean Squared Error (MSE) with one-hot encoded targets |
| Optimizer | AdamW |
| Learning Rate (LR) | $5 \times 10^{-4}$ |
| Weight Decay (WD) | 0.0 (for main results), 0.01 (for Appendix C comparison) |
| AdamW $\beta_1$ | 0.9 (PyTorch default) |
| AdamW $\beta_2$ | 0.999 (PyTorch default) |
| AdamW $\epsilon$ | $10^{-8}$ (PyTorch default) |
| Optimization Steps | $10^7$ |
| Data Type (PyTorch) | 'torch.float64' |
| Random Seed | 0 (for all libraries) |
| Software Framework | PyTorch |
| `HTSR` Tool | WeightWatcher v0.7.5.5 [8] |

**Note on Weight Decay:** The primary results presented in this paper, particularly those demonstrating grokking followed by late-stage generalization collapse (Figure 1), were obtained with weight decay explicitly set to 0. This allows observation of the learning dynamics driven purely by the optimizer and the loss landscape while exhibiting both phenomena, whereas the other proposed measures fail to detect the grokking transition of increasing test accuracy. Runs with non-zero weight decay (e.g., WD=0.01, see Appendix C) were also performed for comparison, showing different dynamics but confirming the general utility of `HTSR`.

## B  Comparative Grokking Progress Metrics and Measures

**Weight Norm Analysis**  Following observations that weight decay can influence grokking [6], we monitor the $l^2$ norm of the network's weights,

$$||\mathbf{W}||_2 = \sqrt{\sum_l ||\mathbf{W}_l||_F^2}, \tag{7}$$

throughout training. We specifically run experiments with weight decay disabled (WD=0) to isolate the effect of the optimization dynamics on the norm itself.

**Activation Sparsity.** For a given layer with activations $b_{i,j}$ (representing the activation of neuron $j$ for input example $i$), the activation sparsity $A_s$ is defined as:

$$A_s = \frac{1}{T} \sum_{i=1}^{T} \frac{1}{n} \sum_{j=1}^{n} \mathbf{1}(b_{i,j} < \tau), \tag{8}$$

where $T$ is the number of training examples, $n$ is the number of neurons in the layer, $\tau$ is a chosen threshold, and $\mathbf{1}(\cdot)$ is the indicator function. This metric measures neuron inactivity. Prior studies have linked activation sparsity to generalization [5, 12, 14] and reported specific dynamics such as plateauing before grokking [2] or an increase preceding a rise in test loss [3].

**Absolute Weight Entropy.** For a weight matrix $W \in \mathbb{R}^{m \times n}$, the absolute weight entropy $H_{abs}(W)$ is given by:

$$H_{abs}(W) = -\sum_{i=1}^{m} \sum_{j=1}^{n} |w_{i,j}| \log |w_{i,j}|. \tag{9}$$

This entropy quantifies the spread of absolute weight magnitudes. Golechha et al. [2] suggested its sharp decrease signals generalization.

**Approximate Local Circuit Complexity.** Let $L^{(W)}(x)$ denote the output logits for input $x$ using weights $W$, and let $L^{(W')}(x)$ denote the logits when 10% of the weights are set to zero (forming $W'$). The approximate local circuit complexity, denoted $\Lambda_{LC}$, is the summed KL divergence:

$$\Lambda_{LC} = \sum_{k=1}^{N_{data}} \sum_{j \in \mathcal{C}} \Pr\big(j|L^{(W)}(x_k)\big) \log \frac{\Pr\big(j|L^{(W)}(x_k)\big)}{\Pr\big(j|L^{(W')}(x_k)\big)}. \tag{10}$$

Here, $N_{data}$ is the number of training examples $x_k$, $\mathcal{C}$ is the set of classes, and $\Pr(j|L(x))$ is the probability of class $j$ derived from the logits $L(x)$ (e.g., via softmax). This measure captures output sensitivity to minor weight perturbations. Lower $\Lambda_{LC}$ has been linked to stable, generalizable representations [2].

# C   Experiment with Weight Decay

To further understand the influence of weight decay on the observed generalization dynamics and the behavior of our tracked metrics, we conducted an experiment identical to our main study (WD=0) but with a small amount of weight decay (WD=0.01) applied. The training curves and metric evolutions for this WD=0.01 experiment are presented in Figures 6, and 7.

A key characteristic of training with weight decay is the tendency for the $l^2$ norm of the weights to decrease over time, or stabilize at a lower value, which is observed in this experiment (Figure 7). This contrasts with the continuously increasing $l^2$ weight norm seen in our primary WD=0 experiments.

In this WD=0.01 regime, the network still achieves a high level of test accuracy. Notably, after the initial grokking phase, the test accuracy slightly decreases and then enters a prolonged plateau, maintaining near peak performance for a significant number of optimization steps (Figure 6). Correspondingly, the average heavy-tail exponent, $\alpha$, also exhibits the decrease and a distinct plateau around the critical value of $\alpha \approx 2$ during this period (Figure 6, top left panel).

The other progress measures considered—Activation Sparsity and Approximate Local Circuit Complexity—also tend to plateau or stabilize during this phase of peak test performance in the WD=0.01 setting (Figure 7). This contrasts with the WD=0 scenario where, despite eventual grokking, the system does not find such a stable long-term plateau and instead proceeds towards a late-stage generalization collapse. The observation that $\alpha$ (and other metrics) plateau in conjunction with peak, stable test accuracy under traditional weight decay settings aligns with some existing understanding of well-regularized training.

While HTSR and the $\alpha$ exponent provide valuable insights in both regimes, its unique capability to signal impending collapse in the absence of weight decay underscores its importance for understanding layer dynamics under various scenarios.

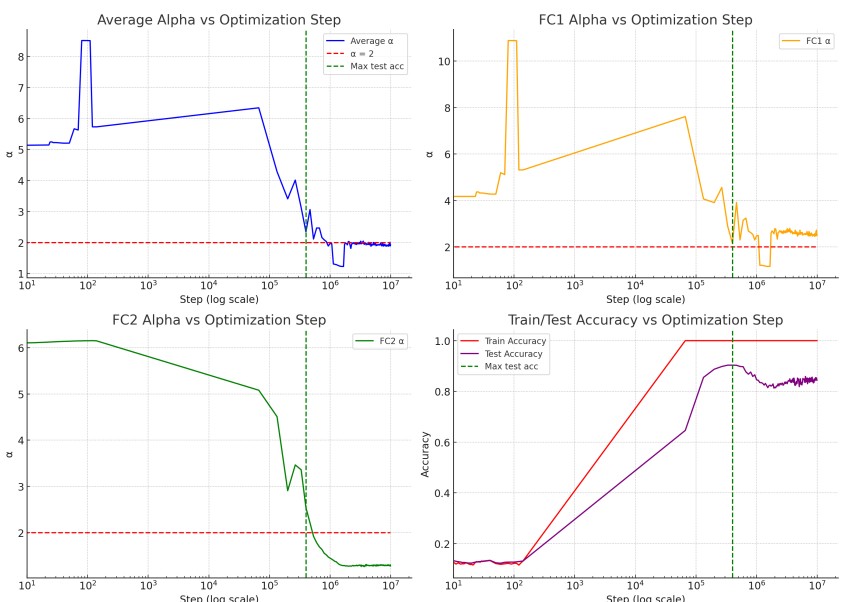

Figure 6: HTSR $\alpha$ exponent evolution for the MLP trained with WD=0.01.

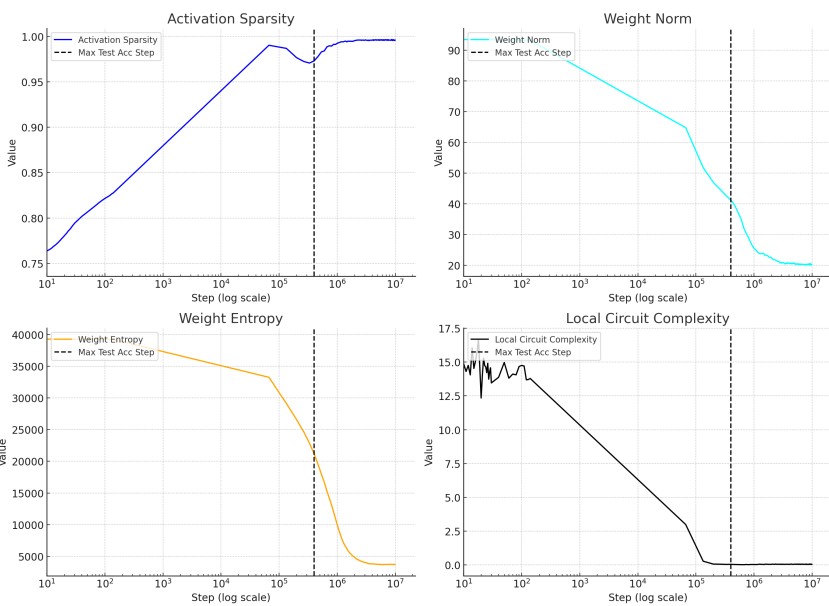

Figure 7: Progress measures (Activation Sparsity, Weight Entropy, Circuit Complexity) and $l^2$ Weight Norm for the MLP trained with WD=0.01.

## D   Statistical Analysis and Validation of Correlation Traps

Here, to further validate the presence of correlation traps for the zero weight decay $WD = 0$ experiment , we report the results of statistical tests designed to determine if the randomized ESD of the $\mathbf{W}^{rand}$ fits an MP distribution or not. Briefly we fit the ESD to a MP distribution and report the fitted variance $\sigma_{mp}$, the Kolmogorov-Smirnov (KS) statistic of the fit, and the p-value for the MP fit as the null model. We also report the number of correlation traps, as determined using the open-source WeightWatcher tool[8]. Results for layer FC1 are presented in Table 4. Results for FC2 are similar (not shown). Additional details are provided in the supplementary material.

Table 4: **Statistical validation of correlation traps.** Selected results for layer FC1 at different training stages for zero weight decay ($WD = 0$) experiment. MP Variance ($\sigma_{MP}$) Kolmogorov-Smirnov (KS) test statistic, p-value for MP fit, and number of detected correlation traps. Pre-grokking $\sim 10^5$ steps, Grokking $10^6$ steps, and Anti-grokking $10^7$ steps,

| Model State | MP variance ($\sigma_{mp}$) | KS Statistic | p-value | # Traps |
|---|---|---|---|---|
| Pre-Grokking | $\approx 1.002$ | 0.0120 | $\approx 1.0$ | 0 |
| Grokking (Max Test Accuracy) | $\approx 0.999$ | 0.0212 | $\approx 1.0$ | 0 |
| Anti-Grokking (Collapse) | $\approx 0.949$ | 0.3044 | $1.877 \times 10^{-5}$ | 9 |

**Initial Layer State (Pre-Grokking WD=0):**    Immediately after initialization, the network weights are expected to be largely random, and their ESD should conform well to the MP distribution. Figure 2 (Right) shows an MP fit to an ESD from a representative layer $\mathbf{W}^{rand}$ of the newly initialized model. A KS test comparing this empirical ESD to the fitted MP distribution (using $\sigma_{mp} \approx 1.0024$ as estimated by `WeightWatcher`) yielded a KS statistic of $0.0120$ and a p-value $\approx 1.0$. This high p-value indicates this ESD is statistically consistent with the MP distribution, as expected.

**Best Layer State (Grokking phase WD=0):**    As the network learns and reaches its maximum test accuracy, significant structure develops in the elements of the weight matrices $W_{i,j}$. This can be seen by randomizing the layer weight matrix elementwise, $\mathbf{W} \rightarrow \mathbf{W}^{rand}$, and plotting ESD, and looking for deviations from the theoretical MP distribution. The ESD now typically exhibits a pronounced heavy tail, with eigenvalues extending beyond the bulk region that might be approximated by an MP fit. For our model at peak test accuracy, the KS test against a fitted MP model ($\sigma_{mp} \approx 0.999$) resulted in a KS statistic of $0.0212$ and a p-value $\approx 1$. Again, this is an MP distribution.

**Final Layer State (Anti-Grokking phase WD=0):**    In the late-stage of training, as the model undergoes generalization collapse and enters an over-correlated state (characterized by $\alpha < 2$), the ESD of $\mathbf{W}^{rand}$ structure continues to reflect a non-random configuration. The KS test for the final model against an MP fit (with an estimated $\sigma_{mp} \approx 2$) yielded a KS statistic of $0.3044$ and a p-value of $1.877 \times 10^{-5}$ Figure 3 (Right) . This result further confirms that the network's structure remains significantly different from a random matrix baseline, consistent with the highly correlated or near rank-collapsed state indicated by our `HTSR` analysis.

These quantitative comparisons demonstrate a transition from an initially random-like state (consistent with MPD) to progressively more structured, non-random states as learning occurs and eventually leads to over-correlation. The inability of the MP distribution to describe these learned features, especially the heavy tails, necessitates the use of tools like the `HTSR` theory, the PL exponent $\alpha$, and the open-source `WeightWatcher` tool, to properly characterize these complex correlation structures and their relationship to generalization performance.

# NeurIPS Paper Checklist

The checklist is designed to encourage best practices for responsible machine learning research, addressing issues of reproducibility, transparency, research ethics, and societal impact. Do not remove the checklist: **The papers not including the checklist will be desk rejected.** The checklist should follow the references and follow the (optional) supplemental material. The checklist does NOT count towards the page limit.

Please read the checklist guidelines carefully for information on how to answer these questions. For each question in the checklist:

- You should answer [Yes] , [No] , or [NA] .
- [NA] means either that the question is Not Applicable for that particular paper or the relevant information is Not Available.
- Please provide a short (1–2 sentence) justification right after your answer (even for NA).

**The checklist answers are an integral part of your paper submission.** They are visible to the reviewers, area chairs, senior area chairs, and ethics reviewers. You will be asked to also include it (after eventual revisions) with the final version of your paper, and its final version will be published with the paper.

The reviewers of your paper will be asked to use the checklist as one of the factors in their evaluation. While "[Yes] " is generally preferable to "[No] ", it is perfectly acceptable to answer "[No] " provided a proper justification is given (e.g., "error bars are not reported because it would be too computationally expensive" or "we were unable to find the license for the dataset we used"). In general, answering "[No] " or "[NA] " is not grounds for rejection. While the questions are phrased in a binary way, we acknowledge that the true answer is often more nuanced, so please just use your best judgment and write a justification to elaborate. All supporting evidence can appear either in the main paper or the supplemental material, provided in appendix. If you answer [Yes] to a question, in the justification please point to the section(s) where related material for the question can be found.

IMPORTANT, please:

- **Delete this instruction block, but keep the section heading "NeurIPS Paper Checklist",**
- **Keep the checklist subsection headings, questions/answers and guidelines below.**
- **Do not modify the questions and only use the provided macros for your answers**.

