# OpenReview forum: "Grokking and Generalization Collapse: Insights from HTSR theory"
_NeurIPS.cc/2025/Conference — Submitted to NeurIPS 2025_

### Official Review · Reviewer_LTrn · 2025-06-10

**Clarity:** 4
**Significance:** 3
**Originality:** 3
**Rating:** 4
**Confidence:** 4

**Summary:**

This paper investigates the phenomena of Grokking and Generalization Collapse (Anti-Grokking) from the perspective of HTSR theory. The authors find that the parameter $\alpha$ in the HTSR theory, which characterizes the empirical spectral distribution (ESD) of the parameter matrix’s eigenvalues, can predict both grokking and anti-grokking behavior. They further propose the concept of *Correlation Traps*—abnormally high eigenvalues that lie outside the MP (Marchenko–Pastur) fitting curve after element-wise random shuffling of the parameter matrix—which are considered to be the cause of anti-grokking. The proposed indicators show stronger correlation with grokking than existing metrics such as weight norm, weight entropy, and activation sparsity.

**Questions:**

1. Why does a lower $\alpha$ indicate stronger fitting ability to the training set? Could analyses of gradient or parameter matrix singularity—such as those based on Neural Tangent Kernel (NTK) theory—help in understanding this relationship?
2. Why do outliers emerge outside the Marchenko–Pastur (MP) bulk after element-wise random shuffling of parameter matrices in the overfitting phase? Could this phenomenon be theoretically justified by making assumptions about the singular values of the original parameter matrix?

I would be happy to increase my score to 4 or 5 if authors could kindly provide more theoretical insights.

**Ethical Concerns:**

["NO or VERY MINOR ethics concerns only"]

**Final Justification:**

I appreciate the authors’ detailed responses. As also noted by other reviewers, one of the key limitations of this work is the lack of a deeper interpretation of the underlying relationship between grokking and the HTSR metric. While the authors have not yet provided a rigorous theoretical justification for this connection, their empirical analysis and theoretical reasoning regarding the effects of entry-wise shuffling of weight matrices seems make sense to me.

Therefore, despite the current limitations, I am willing to raise my score to 4 to encourage such essential, fundamental work, which has the potential to inspire future work with more in-depth theoretical and empirical investigations. I'd like to leave the final decision to AC.

**Limitations:**

yes

**Paper Formatting Concerns:**

1. The conclusion section is too long, making it look more like an introduction.
2. In line 57, did you mean *instability*?
3. In line 121, did you mean the start of the *PL* tail?
4. In the caption of Figure 4, what do "top," "middle," and "bottom" refer to specifically?

**Quality:**

3

**Strengths And Weaknesses:**

**Strengths:**
1. The authors introduce the phenomenon of anti-grokking and use the $\alpha$ parameter and correlation traps for its prediction and explanation, offering a novel perspective to existing work on grokking.
2. The paper clearly defines the mathematical tools used and explains them with helpful figures, making the presentation easy to follow.
3. The experimental results support the claim that $\alpha$ and correlation traps can predict grokking and anti-grokking.

**Weaknesses:**
1. The paper offers only empirical observations and lacks theoretical depth. Although correlation traps are proposed to explain anti-grokking, the evidence is limited to empirical correlation, without theoretical justification. Please refer to the *Questions* section for further discussion.

---

> ### Author Rebuttal · Authors · 2025-07-30
>
> # QN1 : why does a lower alpha work better
>
> ### Why does a smaller spectral exponent $\alpha$ imply a better fit to the training set?
>
> In the **SETOL** framework&nbsp;[Martin & Hinrichs 2025] one finds that the generalization accuracy or quality $\mathcal{Q}$ of a layer can be expressed as a sum of integrated cumulants of the layer weight matrix $\mathbf{W}$
>
> $$
> W \\in \\mathbb{R}^{M\\times N},
> $$
>
>
> $$\mathcal{Q^2}=\sum_{i}\int_{\lambda_{min}}^{\lambda_i}R(z)dz$$, where $\lambda$ are the tail eigenvalues of the layer correlation matrix
>
> $$ \mathbf{X}=\tfrac{1}{N}\mathbf{W}^{\top}\mathbf{W}$$
>
> Where the eigenvalues are fit to a power law distribution with exponent $\alpha$
>
> $$\rho(\lambda)\propto\lambda^{-\alpha}, \lambda_i < \lambda_{min}$$  As $\alpha$ reaches the lower bound of 2 the cumulants $R(z)$ are larger and therefore the quality $\mathcal{Q}$ is larger.
>
> The lower bound of $\alpha=2$ is associated with a phase boundary between perfect generalization and overfitting. This is described using Wilsons Exact renormalization group theory in SETOL.
>
> When $\alpha<2$ and/or there are correlation traps then $\mathbf{W}$ is atypical. As with any statistical sampling method you cannot sample from an atypical distribution (ie: Cauchy,Levy etc) where the moments do not exist/diverge and obtain reliable out of sample predictions , and this is why the test error diverges.
>
> The full derivation exists at reference [9] with an updated version on Arxiv.
>
>
> ---
>
>
> ## Conditions for outliers after Entry‑wise Shuffling of a Weight Matrix (qn2 Reviewer LTrn)
> As shown in reference [10]. If we randomize the weight matrix $\mathbf{W}$ element-wise it becomes IID and any remaining spikes are indicative of large rank-1 perturbations (which are called correlation traps ref [9]).
>
> We review the proof here and are happy to include an exposition of this form in the appendix and/or the supplementary material to answer the reviewers question.
>
> **Neural‑network layer setting.**
> Consider a layer weight matrix
>
> $$
> W \\in \\mathbb{R}^{M\\times N},
> $$
>
> mapping an \(N\)-dimensional input to an \(M\)-dimensional output.  Define
>
> $$
> \\gamma := \\frac{M}{N} \\in (0,\\infty).
> $$
>
> ---
>
> ## Notation
>
> For any matrix \(W\) set
>
> $$
> X(W) := \\frac{1}{N} W^{\\!\\top}W,
> \\qquad
> \\lambda_{\\max}(W) := \\lambda_{\\max}\\bigl(X(W)\\bigr).
> $$
>
> Fix a variance proxy \\(\\sigma^{2}>0\\) and the Marchenko–Pastur edges
>
> $$
> \\lambda_{-} := \\sigma^{2}\\bigl(1-\\sqrt{\\gamma}\\bigr)^{2},
> \\qquad
> \\lambda_{+} := \\sigma^{2}\\bigl(1+\\sqrt{\\gamma}\\bigr)^{2}.
> $$
>
> ---
>
> ## Limit convention
>
> As is typical with MP limits We let \\(N\\to\\infty\\) with \\(M/N\\to\\gamma\\); every limit below is along this sequence.
>
> \\(N\\gg1\\)
>
> ---
>
> ## Norm conventions
>
> For \\(x\\in\\mathbb{R}^{N}\\), \\(\\lVert x\\rVert_{2}\\) denotes the Euclidean norm;
> for \\(A\\in\\mathbb{R}^{M\\times N}\\), \\(\\lVert A\\rVert\\) is the spectral norm;
> for symmetric \\(S\\), \\(\\lambda_{\\max}(S)\\) is its largest eigenvalue.
>
> ---
>
> ## Marchenko–Pastur reminder
>
> The density of \\(\\mathrm{MP}_{\\sigma^{2},\\gamma}\\) is
>
> $$
> \\rho_{\\mathrm{MP}}(\\lambda)
> = \\frac{\\sqrt{(\\lambda_{+}-\\lambda)(\\lambda-\\lambda_{-})}}
>        {2\\pi\\,\\sigma^{2}\\,\\gamma\\,\\lambda}
>   1_{[\\lambda_{-},\\lambda_{+}]}(\\lambda).
> $$
>
> ---
>
> ## Element‑wise shuffling (scrambling)
>
> Let
>
> $$
> \\pi:\\{1,\\dots,M\\}\\times\\{1,\\dots,N\\}
> \\longrightarrow
> \\{1,\\dots,M\\}\\times\\{1,\\dots,N\\}
> $$
>
> be any permutation.  The scrambled weight matrix is
>
> $$
> W_{\\mathrm{rand}} := (W_{ij})_{\\pi(i,j)}.
> $$
>
> ---
>
> ## Shorthand
>
> Define
>
> $$
> A_N := X\\bigl(W^{(N)}_{\\mathrm{rand}}\\bigr),
> $$
>
> so that
>
> $$
> \\lambda_{\\max}(A_N)
> = \\lambda_{\\max}\\bigl(X(W^{(N)}_{\\mathrm{rand}})\\bigr).
> $$
>
> ---
>
> ## Definition 1 (typical vs. atypical)
>
> A sequence \\(\\{W^{(N)}\\}\\) is
>
> - **Typical** if
>
>   $$
>   \\lambda_{\\max}(A_N)
>   \\xrightarrow[N\\to\\infty]{} \\lambda_{+}.
>   $$
>
> - **Atypical** if
>
>   $$
>   \\liminf_{N\\to\\infty}
>   \\bigl[\\lambda_{\\max}(A_N)-\\lambda_{+}\\bigr] > 0.
>   $$
>
> ---
>
> ## Theorem 2 (single‑entry sufficient condition)
>
> Assume \\(M/N\\to\\gamma\\in(0,\\infty)\\) and that for some index pair \\((p_N,q_N)\\) we have
>
> $$
> \\frac{|W^{(N)}_{p_N q_N}|}{\\sqrt{N}}
> \\xrightarrow[N\\to\\infty]{} \\theta,
> \\qquad
> \\theta > \\sigma\\bigl(1+\\sqrt{\\gamma}\\bigr).
> $$
>
> Then there exists \(N_0\) such that for all \(N\\ge N_0\),
>
> $$
> \\lambda_{\\max}(A_N) > \\lambda_{+},
> $$
>
> so the layer is **atypical**.
>
> ### Proof
>
> Let
>
> $$
> a_N := \\max_{i,j} |W_{ij}|.
> $$
>
> Because scrambling is a permutation of entries, the value of \\(a_N\\) is unchanged by it.
> Choose a column in \\(W_{\\mathrm{rand}}\\) that contains an entry of magnitude \\(a_N\\).
> Denote by \\(e_{c_N}\\in\\mathbb{R}^{N}\\) the corresponding standard basis vector (with a 1 in position \\(c_N\\) and zeros elsewhere).
> Since \\(\\lVert e_{c_N}\\rVert_2 = 1\\), the Rayleigh quotient yields
>
> $$
> e_{c_N}^{\\!\\top} A_N \, e_{c_N}
>   = \\frac{1}{N}\\,\\lVert W_{\\mathrm{rand}} e_{c_N}\\rVert_{2}^{2}
>   \\;\\ge\\; \\frac{a_N^{2}}{N}.
> $$
>
> By the Courant–Fischer principle, for any symmetric matrix \(S\),
>
> $$
> \\lambda_{\\max}(S)
>   = \\max_{\\lVert x\\rVert_2 = 1} x^{\\!\\top} S x,
> $$
>
> hence
>
> $$
> \\lambda_{\\max}(A_N) \\;\\ge\\; \\frac{a_N^{2}}{N}.
> $$
>
> By initial assumption
>
> $$
> a_N \\ge |W^{(N)}_{p_N q_N}| = (\\theta + o(1))\\sqrt{N},
> $$
>
> so that
>
> $$
> \\frac{a_N^{2}}{N} \\xrightarrow[N\\to\\infty]{} \\theta^{2}.
> $$
>
> Because \\(\\theta > \\sigma(1+\\sqrt{\\gamma})\\), we have
>
> $$
> \\theta^{2} > \\sigma^{2}(1+\\sqrt{\\gamma})^{2} = \\lambda_{+}.
> $$
>
> Consequently, there exists \(N_0\\) such that for all \(N\\ge N_0\),
>
> $$
> \\lambda_{\\max}(A_N) \\;>\\; \\lambda_{+}.
> $$
>
> Therefore the layer under this condition is **atypical**. \\(\\square\\)
>
> ---
>
> ## Connection with the BBP transition
>
> In the classic spiked‑covariance model \\(W=Z+\\theta\,uv^\\top\\) (rank‑one signal plus i.i.d.\ noise),
> an eigenvalue separates from the Marchenko–Pastur bulk precisely when
>
> $$
> \\theta^{2}=\\sigma^{2}(1+\\sqrt{\\gamma})^{2}.
> $$
>
> The condition above places the scrambled matrix in the super‑critical regime
> \\((\\theta^{2} > \\sigma^{2}(1+\\sqrt{\\gamma})^{2})\\),
> so an outlier survives even after correlations are destroyed.
>
> ---
>
> ### Discussion and intuition
>
> Modern empirical work suggests that *stochastic gradient descent* (SGD) can drive a handful of coordinates of a weight matrix toward abnormally large magnitudes, producing either rank‑one–like spikes or genuinely heavy‑tailed weight distributions [2]. Such spikes distort the spectrum of the layer’s Gram matrix; eigenvalues well above the MP bulk appear precisely when catastrophic drops in test accuracy are observed.
>
> Entry‑wise shuffling deliberately *destroys all remaining correlations* in the weight matrix while preserving the multiset of magnitudes. If a large eigenvalue survives the shuffle, there must be a *pure‑magnitude* signal, rendering the layer atypical.
>
> Theorem 2 and BBP shows that the presence of even one entry growing such that
> \\((\\theta^{2} > \\sigma^{2}(1+\\sqrt{\\gamma})^{2})\\),
>
> suffices to guarantee such a surviving outlier, confirming the empirical link between unusually large coordinates and atypical spectra.
>
>
> This is also highlighted in Reference [3], see Section 5.3 (Bulk+Spikes).
>
>
> ## References
>
> 1. J. Baik, G. Ben Arous & S. Péché, *Ann. Probab.* 33 (2005), 1643–1697.
> 2. M. Gürbüzbalaban, U. Şimşekli & L. Zhu, *Proc.* 38th ICML (2021), 3964–3975.
> 3. Charles H. Martin and Michael W. Mahoney. Implicit self-regularization in deep neural networks: Evidence from heavy-tailed spectral analysis, 2021. (JMLR)
> 4. Charles H. Martin and Christopher Hinrichs SETOL: A Semi-Empirical Theory of (Deep) Learning, 2025 (Arxiv 2507.17912)

---

> > ### Comment · Reviewer_LTrn · 2025-08-02
> >
> > I appreciate the authors’ detailed responses. As also noted by other reviewers, one of the key limitations of this work is the lack of a deeper interpretation of the underlying relationship between grokking and the HTSR metric. While the authors have not yet provided a rigorous theoretical justification for this connection, their empirical analysis and theoretical reasoning regarding the effects of entry-wise shuffling of weight matrices seems make sense to me.
> >
> > Therefore, despite the current limitations, I am willing to raise my score to 4 to encourage such essential, fundamental work, which has the potential to inspire future work with more in-depth theoretical and empirical investigations. I'd like to leave the final decision to AC.

---

### Official Review · Reviewer_RbVm · 2025-06-30

**Clarity:** 3
**Significance:** 3
**Originality:** 3
**Rating:** 4
**Confidence:** 2

**Summary:**

The authors study the relationship between generalization and the power-law exponent $\alpha$ in the empirical spectral density (ESD) of network weight matrices in a single setting where grokking occurs. HTSR theory predicts that $\alpha = 2$ is the ideal value for networks to generalize. When training 3-layer ReLU MLPs on MNIST without weight decay (the authors scale the weights by 8 to induce grokking, as was introduced in (Liu et al. "Towards Understanding Grokking..." and "Omnigrok..."), the authors find that their network's generalization eventually improves after a prolonged period of training, but then worsens with continued training. They find that the era of training where $\alpha \approx 2$ roughly corresponds with the maximum test accuracy, suggesting that HTSR theory can be applied to understand grokking.

**Questions:**

* In my view, in order for this article to warrant acceptance to the conference main track, it would have to more thoroughly investigate the relationship between the $\alpha$ and generalization in grokking. In particular, the authors should conduct experiments where various parameters are varied that affect the timing of generalization in grokking, and then see whether the $\alpha$ dynamics also vary in the way we'd expect. Some suggestions:
  - Change the large initialization scale of the parameters in the network to vary grokking timing. Instead of scaling the weights and biases by 8, one could vary this scaling factor, which might change the timing of generalization in the networks. Does $\alpha$ approach $\approx 2$ when test accuracy is maximized across each choice of initialization scale?
  - Sweep the weight decay parameter. In Liu et al. (2022) (Omnigrok), this directly changes the timing of generalization in their MNIST networks. Does the timing at which $\alpha$ reach the optimal region $\approx 2$ change accordingly?
  - Other networks. The easiest would be to study small transformers or MLPs on algorithmic tasks like modular addition.

**Ethical Concerns:**

["NO or VERY MINOR ethics concerns only"]

**Final Justification:**

I've raised my score to borderline accept now that the authors have run experiments on modular addition and on MNIST with a different weight initialization scale, and found that the HTSR $\alpha \approx 2$ correlates with grokking in these settings too. This seems promising! Note however that I've lowed my confidence to a 2, since I I'm not familiar with HTSR theory. The authors say they'll add the new results to the appendix, though I'd suggest rewriting the paper to put the results across tasks into the main body of the paper. They seem to meaningfully strengthen the paper's core claim.

**Limitations:**

Yes. The authors mention the main limitation I identified above, that their experiments are only on a single setup.

**Quality:**

2

**Strengths And Weaknesses:**

### Strengths
The main result of the paper is interesting: the dynamics of $\alpha$ throughout training correspond with how the network generalizes in the manner that HTSR theory would predict. This is an interesting proof of concept, and warrants further investigation.

### Weaknesses
* The relationship between $\alpha$ and generalization is only tested in a single setting. While in Appendix C the authors also test the case where weight decay is nonzero, they do not study the relationship between $\alpha$ and grokking on any other tasks where grokking occurs, like the algorithmic tasks where grokking was initially observed. While this result is interesting, it is difficult to tell whether this finding is robust.
* Numerous small misspellings and grammatical and formatting errors that make the paper feel thrown-together. The most surprising is that the caption for Figure 4 includes "Top", "Middle", and "Bottom" descriptions for subfigures that don't correspond with the actual arrangement of the subfigures on that plot.

---

> ### Author Rebuttal · Authors · 2025-07-30
>
> # Response to RbVm
> We thank the reviewer for the constructive feedback and now try to answer some of these questions , beginning with the modular‑addition study (the new experiment whose run‑time exceeded limits by a month) and concluding with the changed initialization‑scale experiment requested.
>
> TLDR 1. For a change in the initialization scale (from 8 to 4) , HTSR alpha tracks the generalization.
> 2. We did the modular addition experiment and also observed the antigrokking phase and can detect it using HTSR theory (alpha !=2 and correlation traps>0). We are happy to add these results to the appendix or the supplementary, and hope that this addresses the concerns by the reviewer .
>
> ---
>
> ## 1 Modular‑addition transformer Experiment
>
> ###  Why this task and why it took time
>
> Modular addition (predicting $x+y\bmod P$) was chosen because it was the original algorithmic setting in which grokking occured. Running for $10^{7}$ steps on a single Quadro P2000 (5 GB) unfortunately went on to roughly one month longer than initially budgeted...here are the results (which has been now added to the appendix)
>
>
>
> Prime chosen = 113
> ###  Model architecture
>
> | Hyper‑parameter                    | Value                            |
> | ---------------------------------- | -------------------------------- |
> | Layers                             | **1**                            |
> | Model dimension $d_{\text{model}}$ | **128**                          |
> | MLP hidden size $d_{\text{mlp}}$   | **512**                          |
> | Heads × head dim                   | **4 × 32**                       |
> | Context length $n_{\text{ctx}}$    | **3**                            |
> | Activation                         | **ReLU**                         |
> | LayerNorm                          | **disabled** *(use\_ln =False)*  |
> | Vocabulary size                    | **114** (= equals\_token + 1)    |
>
>
>
> ### HTSR results (α)
>
> | Layer                   | Pre‑grok α      | Grok α          | Anti‑grok α     |
> | ----------------------- | --------------- | --------------- | --------------- |
> | embed.embed             | 2.81            | **1.92**        | 3.57            |
> | blocks.0.attn.W\_Q      | 4.04            | **2.86**        | 3.17            |
> | blocks.0.attn.W\_K      | 3.99            | **2.00**        | 3.20            |
> | blocks.0.attn.W\_V      | 4.66            | **1.56**        | 3.87            |
> | blocks.0.attn.out\_proj | 3.85            | **1.57**        | 4.04            |
> | blocks.0.mlp.fc1        | 4.99            | **1.71**        | 4.74            |
> | blocks.0.mlp.fc2        | 5.22            | **2.17**        | 5.03            |
> | unembed.unembed         | 3.27            | **2.37**        | 3.53            |
> | **Mean ± Std**          | **4.10 ± 0.83** | **2.02 ± 0.44** | **3.89 ± 0.68** |
>
> ### 1.6 Accuracy trajectories
>
> | Phase     | Train Acc.      | Test Acc.           |
> | --------- | --------------- | ------------------- |
> | Pre‑grok  | 1.0000 ± 0.0000 | 0.4056 ± 0.2678     |
> | Grok      | 1.0000 ± 0.0000 | **0.9725 ± 0.0166** |
> | Anti‑grok | 1.0000 ± 0.0000 | 0.5978 ± 0.1116     |
>
> Correlation trap analysis is also performed and the table is given below
>
> | Layer                   | Pre‑grok α      | Grok α          | Anti‑grok α     |
> | ----------------------- | --------------- | --------------- | --------------- |
> | embed.embed             | 0.00            | 0.00            | 3.48            |
> | blocks.0.attn.W\_Q      | 0.00            | 0.00            | 3.74            |
> | blocks.0.attn.W\_K      | 0.00            | 0.00            | 3.72            |
> | blocks.0.attn.W\_V      | 0.00            | 0.00            | 2.91            |
> | blocks.0.attn.out\_proj | 0.00            | 0.00            | 3.03            |
> | blocks.0.mlp.fc1        | 0.00            | 0.00            | 5.26            |
> | blocks.0.mlp.fc2        | 0.00            | 0.00            | 6.05            |
> | unembed.unembed         | 0.00            | 0.00            | 4.83            |
> | **Mean ± Std**          | **0.00 ± 0.00** | **0.00 ± 0.00** | **4.13 ± 1.13** |
>
>
> As predicted by HTSR, the global mean exponent $\alpha$ gravitates tightly around 2 during the test‑accuracy peak and deviates from 2 thereafter. Also as we mention in the paper we see the increase of the outlier eigen values from the marchenko pastur bulk in all layers during the antigrok phase.
>
>
> ## 2 Initialization‑scale = 4 ablation (MNIST MLP)
>
> This next experiment directly answers the reviewer’s first suggestion—*“Change the large initialization scale … does α approach 2 when test accuracy is maximized?”*—while keeping every other setting identical to the baseline described in experimental section of the paper.
>
> | Layer | Pregrok (10²–5×10⁴) mean | Grok (5×10⁴–5×10⁵) mean | Antigrok (>5×10⁵) mean |
> | :----------: | -----------------------: | ----------------------: | ---------------------: |
> | Layer 1 | 8.88 | 2.59 | 1.23 |
> | Layer 2 | 3.84 | 2.77 | 1.49 |
> | **Mean±Std** | **6.36 ± 2.52** | **2.68 ± 0.09** | **1.36 ± 0.13** |
>
>
> | Phase | Train Accuracy (mean ± std) | Test Accuracy (mean ± std) |
> | :------- | --------------------------: | -------------------------: |
> | Pre‑grok | 0.7298 ± 0.0408 | 0.4926 ± 0.0187 |
> | Grok | 1.0000 ± 0.0000 | 0.8879 ± 0.0051 |
> | Antigrok | 1.0000 ± 0.0000 | 0.6184 ± 0.1086 |
>
>
>
> #### correlation trap analysis
> | Layer          | Pre‑grok α      | Grok α          | Anti‑grok α     |
> | -------------- | --------------- | --------------- | --------------- |
> | 1              | 0.00            | 0.00            | 7.00            |
> | 3              | 0.00            | 0.00            | 1.17            |
> | 5              | 0.00            | 0.00            | 5.50            |
> | **Mean ± Std** | **0.00 ± 0.00** | **0.00 ± 0.00** | **4.56 ± 3.03** |
>
>
> here we see that modulating the initialization scale as suggested modulates the timing of generalization in grokking , with the HTSR exponent alpha following this speed up along with the decay that occurs after.
> We also see that our Correlation trap study is robust to the shift in initialization scale.
>
> ---
>
> ### 3 Writing grammatical errors and caption issues
>
> We identified and corrected grammatical and spelling issues (instabili on page 2) (captions on figure 4 with the positions) etc

---

> > ### Comment · Reviewer_RbVm · 2025-08-02
> >
> > Thank you for this response and for running these new experiments. To me, they substantially improve the paper and I've raised my score to 4 but lowered my confidence to 2.

---

### Official Review · Reviewer_pQqz · 2025-07-02

**Clarity:** 3
**Significance:** 2
**Originality:** 2
**Rating:** 4
**Confidence:** 2

**Summary:**

This paper investigates the grokking phenomenon, where under training neural networks generalize suddenly after extensive overfitting, on the task of classification with a 3-layer MLP trained on a 1,000-sample subset of MNIST. The authors identify three distinct phases: an initial “pre-grokking” phase, the familiar grokking generalization phase, and a newly discovered “anti-grokking” phase late in training, where test performance degrades again. They analyze these phases through the lens of Heavy–Tailed Self-Regularization (HTSR) theory, revealing how the heavy-tail power-law (PL) exponent $\alpha$ reflects the transition dynamics between these phases.

**Questions:**

1. To identify the correlation traps, the paper proposes to look into the randomized version $\mathbf{W}^{\mathrm{rand}}$ of the weight matrix and compare it to the MP distribution. Can you provide more details on how the randomization is performed? It seems that the corresponding pipeline is not mentioned in the paper.
2. Inconsistent captions with respect to the figures in Figure 4.
3. What is the difference between the anti-grokking phase and the standard overfitting phenomenon or double descent phenomenon in training neural networks? Are there any tricks, such as early stopping tailored to HTSR signals, weight decay schedules, or data augmentation, that could mitigate the anti-grokking?
4. Why does the experiments only use a small subset of the MNIST dataset as the training dataset? (Far smaller than the test split size) This is somehow strange.

**Ethical Concerns:**

["NO or VERY MINOR ethics concerns only"]

**Final Justification:**

The authors have addressed most of the questions I raised, including the experiment setups and the relationship between different phases. Despite that the new experiments are still relatively toy and simple, I do appreciate the novel idea applying HTSR theory to study the (anti-) grokking phenomenon. I have raised my score accordingly to 4.

**Limitations:**

Please see the Questions part.

**Quality:**

2

**Strengths And Weaknesses:**

**Strengths:**
1. The idea of applying HTSR theory to study the grokking phenomena in training neural networks is interesting and new.
2. The study is able to investigate and signal the grokking phenomenon without weight decay, which previous studies typically relies on, providing new characterizations of such a phenomenon.

**Weaknesses:**
1. The scope of the dataset and architecture is limited to some toy setups. To demonstrate the universality of the proposed method, more experiments on different tasks, datasets, and architectures are necessary.
2. The HTSR theory often relies on simplifying assumptions (e.g., high-dimensional asymptotics). The degree to which these align with the experiments is not deeply probed.

---

> ### Author Rebuttal · Authors · 2025-07-30
>
> # Response to pQqz
> We thank the reviewer for the constructive feedback and now try to answer some of these questions
>
> ## 1 Identification of correlation traps and pipeline for randomization
> The HTSR analysis is conducted using the opensource weightwatcher code as mentioned right below equation 6 in line 151-152 , We set the option randomize='True', The code then randomizes the weight matrix element wise (permutation of the elements).
> (weightwatcher RMT_Util.py line 892)
> Attached is the associated code snippet
> ```
> def permute_matrix(W):
>     """permute a matrix in a reversible way"""
>
>     num_params = np.prod(W.shape)
>     vec = W.reshape(num_params)
>     p_ids = np.random.permutation(np.arange(num_params))
>     p_vec = vec[p_ids]
>     p_W = p_vec.reshape(W.shape)
>
>     return p_W, p_ids
> ```
>
> ## 2 Captions on figure 4
> We have now corrected the captions to better indicate the placement of the actual figures and we thank the reviewer for bringing this to our attention.
>
> ## 3 Antigrokking vs overfitting and mitigating antigrokking
>
> Antigrokking is characterized by one or more atypical weight matrices (one with large eigenvalue outliers which are rank-1 perturbations and or anomalously large elements of the weight matrix) Sampling from this weight matrix can not give good out of sample predictions due to it being atypical. This is analogous to sampling from a pareto distribution/Cauchy distribution which has ill defined moments. (mean variance etc). Pregrokking is regular overfitting where the model simply fails to generalize well. (ie: The layers have not converged yet).
> Antigrokking can be mitigated by clipping the large weight matrix elements and or rank-1 perturbations in each W thus preventing correlation traps from forming.
> Another way to proceed would be to stop as HTSR PL alpha approaches 2.
>
>
> ## 4 Why 1K on mnist
>
> As shown by reference [6] (Towards Understanding Grokking:
> An Effective Theory of Representation Learning) appendix J  in order to induce grokking on Mnist we must take a smaller subset for training (we took 1K just like they did) and followed the same protocol.
>
>
> ## further experiments
>
> We have also run analysis on the Modular addition (predicting $x+y\bmod P$) experiment  , attached are the results.
> We see the antigrokking phase and we are able to detect this through HTSR alpha and correlation traps using weightwatcher.
>
>
> ### HTSR results (α)
>
> | Layer                   | Pre‑grok α      | Grok α          | Anti‑grok α     |
> | ----------------------- | --------------- | --------------- | --------------- |
> | embed.embed             | 2.81            | **1.92**        | 3.57            |
> | blocks.0.attn.W\_Q      | 4.04            | **2.86**        | 3.17            |
> | blocks.0.attn.W\_K      | 3.99            | **2.00**        | 3.20            |
> | blocks.0.attn.W\_V      | 4.66            | **1.56**        | 3.87            |
> | blocks.0.attn.out\_proj | 3.85            | **1.57**        | 4.04            |
> | blocks.0.mlp.fc1        | 4.99            | **1.71**        | 4.74            |
> | blocks.0.mlp.fc2        | 5.22            | **2.17**        | 5.03            |
> | unembed.unembed         | 3.27            | **2.37**        | 3.53            |
> | **Mean ± Std**          | **4.10 ± 0.83** | **2.02 ± 0.44** | **3.89 ± 0.68** |
>
> ###  Accuracy trajectories
>
> | Phase     | Train Acc.      | Test Acc.           |
> | --------- | --------------- | ------------------- |
> | Pre‑grok  | 1.0000 ± 0.0000 | 0.4056 ± 0.2678     |
> | Grok      | 1.0000 ± 0.0000 | **0.9725 ± 0.0166** |
> | Anti‑grok | 1.0000 ± 0.0000 | 0.5978 ± 0.1116     |
>
> Correlation trap analysis is also performed and the table is given below
>
> | Layer                   | Pre‑grok α      | Grok α          | Anti‑grok α     |
> | ----------------------- | --------------- | --------------- | --------------- |
> | embed.embed             | 0.00            | 0.00            | 3.48            |
> | blocks.0.attn.W\_Q      | 0.00            | 0.00            | 3.74            |
> | blocks.0.attn.W\_K      | 0.00            | 0.00            | 3.72            |
> | blocks.0.attn.W\_V      | 0.00            | 0.00            | 2.91            |
> | blocks.0.attn.out\_proj | 0.00            | 0.00            | 3.03            |
> | blocks.0.mlp.fc1        | 0.00            | 0.00            | 5.26            |
> | blocks.0.mlp.fc2        | 0.00            | 0.00            | 6.05            |
> | unembed.unembed         | 0.00            | 0.00            | 4.83            |
> | **Mean ± Std**          | **0.00 ± 0.00** | **0.00 ± 0.00** | **4.13 ± 1.13** |
>
> ###  Model architecture
>
> | Hyper‑parameter                    | Value                            |
> | ---------------------------------- | -------------------------------- |
> | Layers                             | **1**                            |
> | Model dimension $d_{\text{model}}$ | **128**                          |
> | MLP hidden size $d_{\text{mlp}}$   | **512**                          |
> | Heads × head dim                   | **4 × 32**                       |
> | Context length $n_{\text{ctx}}$    | **3**                            |
> | Activation                         | **ReLU**                         |
> | LayerNorm                          | **disabled** *(use\_ln =False)*  |
> | Vocabulary size                    | **114** (= equals\_token + 1)    |
>
>
>
>
>
> As predicted by HTSR, the global mean exponent $\alpha$ gravitates tightly around 2 during the test‑accuracy peak and deviates from 2 thereafter. Also as we mention in the paper we see the increase of the outlier eigen values from the marchenko pastur bulk in all layers during the antigrok phase.

---

> > ### Comment · Reviewer_pQqz · 2025-08-04
> >
> > The authors have addressed most of the questions I raised. I have raised my score accordingly to 4.

---

### Official Review · Reviewer_vAL9 · 2025-07-02

**Clarity:** 2
**Significance:** 2
**Originality:** 3
**Rating:** 4
**Confidence:** 3

**Summary:**

In this work, the authors study the **Grokking phenomenon** — a behavior in which training accuracy reaches a high level early, while test accuracy remains low until it suddenly improves after many additional training steps. They investigate this using a ReLU network trained on a subset of the MNIST dataset. By significantly extending the number of training steps, they observe a new behavior, which they refer to as **Anti-Grokking**. They further show that the **HTSR** metric serves as a good indicator of all three phases of Grokking behavior.

**Questions:**

### Questions for the Authors

1. Why did you focus only on the MNIST dataset? Are there computational limitations preventing the investigation of this phenomenon on larger or different datasets, or do Grokking and Anti-Grokking behaviors simply not appear as clearly in other domains?

2. Given that the study is limited to MNIST, how can we be confident that the observed relationship between the HTSR value and the three phases of Grokking is **independent of the dataset**?

3. *(Minor)* In **Figure 2**, please specify what each axis represents. This would make the visualization much easier to interpret on its own.

4. As you briefly mentioned in the limitations section, is there any theoretical insight or justification for **why** HTSR serves as a good indicator of the three phases? Even an informal intuition would help strengthen the argument.

**Ethical Concerns:**

["NO or VERY MINOR ethics concerns only"]

**Final Justification:**

A key limitation of this paper is the lack of theoretical results establishing a connection between HTSR and the Grokking phenomenon. As a result, it remains unclear whether the observed patterns reflect a deep relationship or merely a correlation. This makes it difficult to assess the significance of the findings with confidence.
Despite this, the paper proposes a potential relationship between the HTSR metric and Grokking, and the results suggest an interesting and promising direction for further exploration. For that reason, I intend to keep my original score.

That said, I strongly encourage the authors to revise and clean up the paper — particularly by incorporating suggestions from the reviewers and including the new experimental results. One example of needed cleanup is Figure 2, which uses inconsistent fonts: the left side is much larger and bold, whereas the right side is smaller and not bold. Addressing such inconsistencies would help improve the quality and overall presentation of the paper, bringing it to the standard expected at a NeurIPS conference.

**Limitations:**

Yes.

**Paper Formatting Concerns:**

No concern.

**Quality:**

3

**Strengths And Weaknesses:**

I find the relationship between the different training phases and the HTSR value to be very interesting. In particular, showing that other commonly used metrics — such as weight norm or activation sparsity — fail to capture the Anti-Grokking phase is a strong indication that HTSR may be a more informative alternative. That said, since phenomena like Grokking and Anti-Grokking often appear in small or synthetic datasets, there are natural questions about the downstream theoretical and practical relevance of these results. Observing this behavior only on a subset of MNIST makes it difficult to conclude whether the patterns are dataset-independent or more broadly applicable.

---

> ### Author Rebuttal · Authors · 2025-07-30
>
> # Response to vAL9
> We thank the reviewer for the constructive feedback and now try to answer some of these questions
>
>
>
> ##  figure 2
> The axes on Figure two are specified below
> ### Figure 2a axes
> x axis :log eigenvalues
> y axis: log frequency
>
> ### Figure 2b axes
> x axis :log eigenvalues
> y axis: linear frequency
>
> ## Intuition for HTSR
> Most people think about training a neural network as minimizing a loss, but in fact you can think of the layers as coming into equilibrium with the data. In coming to equilibrium you go from a random matrix (at initialization) to a highly correlated structure. The HTSR theory uses Random Matrix theory to quantify how correlated the layer weight matrices $\mathbf{W}$ are. When $\alpha= 2$  they are optimally correlated. If $\alpha\gg 2$ then $\mathbf{W}$ is nearly random and has learned very little information.
>
>  If $\alpha<2$ or has correlation traps then $\mathbf{W}$ is atypical. As with any atypical distribution (ie: Cauchy, Levy) you cannot sample from an atypical  $\mathbf{W}$ and obtain reliable out of sample predictions.
>
> ## 1 Modular‑addition transformer Experiment
>
> ###  Why this task and why it took time
>
> Modular addition (predicting $x+y\bmod P$) was chosen because it was the original algorithmic setting in which grokking occurs. Running for $10^{7}$ steps on a single Quadro P2000 (5 GB) unfortunately went on to roughly one month longer than initially budgeted due to the computational cost...here are the results (which we are happy to add to the appendix and or supplementary material)
>
>
> Prime chosen = 113
> ###  Model architecture
>
> | Hyper‑parameter                    | Value                            |
> | ---------------------------------- | -------------------------------- |
> | Layers                             | **1**                            |
> | Model dimension $d_{\text{model}}$ | **128**                          |
> | MLP hidden size $d_{\text{mlp}}$   | **512**                          |
> | Heads × head dim                   | **4 × 32**                       |
> | Context length $n_{\text{ctx}}$    | **3**                            |
> | Activation                         | **ReLU**                         |
> | LayerNorm                          | **disabled** *(use\_ln =False)*  |
> | Vocabulary size                    | **114** (= equals\_token + 1)    |
>
>
>
> ### HTSR results (α)
>
> | Layer                   | Pre‑grok α      | Grok α          | Anti‑grok α     |
> | ----------------------- | --------------- | --------------- | --------------- |
> | embed.embed             | 2.81            | **1.92**        | 3.57            |
> | blocks.0.attn.W\_Q      | 4.04            | **2.86**        | 3.17            |
> | blocks.0.attn.W\_K      | 3.99            | **2.00**        | 3.20            |
> | blocks.0.attn.W\_V      | 4.66            | **1.56**        | 3.87            |
> | blocks.0.attn.out\_proj | 3.85            | **1.57**        | 4.04            |
> | blocks.0.mlp.fc1        | 4.99            | **1.71**        | 4.74            |
> | blocks.0.mlp.fc2        | 5.22            | **2.17**        | 5.03            |
> | unembed.unembed         | 3.27            | **2.37**        | 3.53            |
> | **Mean ± Std**          | **4.10 ± 0.83** | **2.02 ± 0.44** | **3.89 ± 0.68** |
>
> ### 1.6 Accuracy trajectories
>
> | Phase     | Train Acc.      | Test Acc.           |
> | --------- | --------------- | ------------------- |
> | Pre‑grok  | 1.0000 ± 0.0000 | 0.4056 ± 0.2678     |
> | Grok      | 1.0000 ± 0.0000 | **0.9725 ± 0.0166** |
> | Anti‑grok | 1.0000 ± 0.0000 | 0.5978 ± 0.1116     |
>
> Correlation trap analysis is also performed and the table is given below
>
> | Layer                   | Pre‑grok α      | Grok α          | Anti‑grok α     |
> | ----------------------- | --------------- | --------------- | --------------- |
> | embed.embed             | 0.00            | 0.00            | 3.48            |
> | blocks.0.attn.W\_Q      | 0.00            | 0.00            | 3.74            |
> | blocks.0.attn.W\_K      | 0.00            | 0.00            | 3.72            |
> | blocks.0.attn.W\_V      | 0.00            | 0.00            | 2.91            |
> | blocks.0.attn.out\_proj | 0.00            | 0.00            | 3.03            |
> | blocks.0.mlp.fc1        | 0.00            | 0.00            | 5.26            |
> | blocks.0.mlp.fc2        | 0.00            | 0.00            | 6.05            |
> | unembed.unembed         | 0.00            | 0.00            | 4.83            |
> | **Mean ± Std**          | **0.00 ± 0.00** | **0.00 ± 0.00** | **4.13 ± 1.13** |
>
>
> As predicted by HTSR, the global mean exponent $\alpha$ gravitates tightly around 2 during the test‑accuracy peak and deviates from 2 thereafter. Also as we mention in the paper we see the increase of the outlier eigen values from the Marchenko Pastur bulk in all layers during the Antigrok phase.

---

> > ### Comment · Reviewer_vAL9 · 2025-08-03
> >
> > Thank you for such a great response, and for adding the modular addition result. I agree with the other reviewers that having some theoretical justification for the connection between HTSR and Grokking would be quite interesting. Nevertheless, I believe this work is valuable as it stands, as it paves the way for future research into formalizing this connection.

---

### Official Review · Reviewer_XxTJ · 2025-07-04

**Clarity:** 2
**Significance:** 2
**Originality:** 2
**Rating:** 2
**Confidence:** 3

**Summary:**

The paper contains a numerical study of grokking from the point of view of heavy-tailed self-regularisation (HTSR) theory. This theory posits that the spectral density of weights in a deep neural network is key to understanding grokking, and puts the power-law exponent of this spectrum, $\alpha$, at the centre of these experiments.

The key result of the paper is a strong correlation between three phases of grokking (pre-grokking, grokking, anti-grokking) and the exponent $\alpha$. The main result is that $\alpha$ tracks progress of grokking well and is better than prior progress measures at detecting anti-grokking (i.e. loss of generalisation after extensive training).

**Questions:**

* Why is $\alpha=2$ the "ideal value" that "corresponds to fully optimized layers"? This is a very strong claim with no theoretical support in this paper. If there is a theoretical argument in previous work, this should be explained here as well for completeness.

* The results in this paper are mostly correlational, linking fitted $\alpha$ values with phases of grokking. Is there any evidence that this connection is causal? E.g. can you perturb a given network to have a different $\alpha$ value and see if its train/test accuracy changes accordingly?

* Does this apply to other architectures and regularisation methods?

* What is the role of the Gaussian baseline in page 3? Forgive me if I'm missing something obvious, but it doesn't seem to play any role elsewhere in the paper.

**Ethical Concerns:**

["NO or VERY MINOR ethics concerns only"]

**Final Justification:**

While the new experiments do strengthen the paper, I still have concerns regarding the novelty and theoretical foundations of the work. Although this is a promising empirical evaluation of a novel progress measure, I do not think it meets the standards of NeurIPS.

**Limitations:**

The paper includes a decently-sized Limitations section that covers the key weaknesses of the work. The paper also has several minor typos, formatting errors, and incomplete words and sentences.

**Paper Formatting Concerns:**

Figures are a bit hard to read, especially for colour-blind individuals. Label and legend font sizes are quite small and also difficult to read.

**Quality:**

2

**Strengths And Weaknesses:**

**Strengths**
* The exponent $\alpha$ correlates quite well with grokking. The existence of a threshold makes for a very straightforward stopping criterion.
* The concept of correlation traps is interesting, and the algorithm to detect them (page 5) seems simple and effective. The connection with anti-grokking in Table 2 is quite compelling.

**Weaknessess**
* The results in this paper are exclusively descriptive and correlational. Although HTSR seems to be supported by some prior work (which I admit I'm not familiar with, and is not presented in sufficient depth in the paper), the results presented specifically in this paper do not have any theory behind them. This calls into question where $\alpha$ plays a causal role in grokking or if there is another, more fundamental factor at play.
* The experiments are extremely limited. The paper only explores one particular (and very small) architecture, trained just on MNIST (surprisingly with no mention of the dataset or task in the main text), and with one regularisation method.
* Perhaps the one key claim of the authors (that $\alpha$ tracks generalisation collapse while other metrics don't) is not supported by enough evidence. Judging visually from Fig. 5, it seems that at least some of the measures (e.g. L2) do detect collapse, in the sense that one could very easily put a threshold (e.g. 125) and declare loss of generalisation when L2 exceeds that threshold.
* The literature on grokking and generalisation is growing very quickly, and the Related Work does not do justice to it.

---

> ### Author Rebuttal · Authors · 2025-07-30
>
> We thank the reviewer for the feedback and now try to answer some of these questions
>
> # QN Correlation vs Causation
> To address the reviewers concern about whether HTSR is a cause for generalization.
> In our paper we only claim as in HTSR that $\alpha \approx 2$ is an indicator of generalization, we do not claim that this causes generalization. The HTSR $\alpha$ and Correlation traps are measures and not causes for generalization. Generalization can be thought of as truly a function of the actions the optimizer takes in the parameter manifold, The HTSR theory suggests that as the optimizer moves from the initialization Vector, the layer weights
> $$
> W \\in \\mathbb{R}^{M\\times N},
> $$
> go from random uncorrelated matrices to a highly correlated heavy tail ESD matrices,
>
> ## Marchenko Pastur Baseline comparison
> Also addressing the usage of the gaussian baseline we run KS statistic tests against this baseline for the detection for traps in Table 2 in our paper as seen in Appendix D of the paper titled "Statistical analysis and validation of correlation traps, The table includes the MP variance KS statistic and the P value at different phases
>
> # QN : HTSR alpha theory
>
> ### Why does a smaller spectral exponent $\alpha$ imply a better fit to the training set?
>
> In the HTSR, **SETOL** framework&nbsp;[Martin & Hinrichs 2025] one finds that the generalization accuracy or quality $\mathcal{Q}$ of a layer can be expressed as a sum of integrated cumulants of the layer weight matrix $\mathbf{W}$
>
> $$
> W \\in \\mathbb{R}^{M\\times N},
> $$
>
>
> $$\mathcal{Q^2}=\sum_{i}\int_{\lambda_{min}}^{\lambda_i}R(z)dz$$, where $\lambda$ are the tail eigenvalues of the layer correlation matrix
>
> $$ \mathbf{X}=\tfrac{1}{N}\mathbf{W}^{\top}\mathbf{W}$$
>
> Where the eigenvalues are fit to a power law distribution with exponent $\alpha$
>
> $$\rho(\lambda)\propto\lambda^{-\alpha}, \lambda_i > \lambda_{min}$$  As $\alpha$ reaches the lower bound of 2 the cumulants $R(z)$ are larger and therefore the quality $\mathcal{Q}$ is larger.
>
> The lower bound of $\alpha=2$ is associated with a phase boundary between perfect generalization and overfitting. This is described using Wilsons Exact renormalization group theory in SETOL.
>
> When $\alpha<2$ and/or there are correlation traps then $\mathbf{W}$ is atypical. As with any statistical sampling method you cannot sample from an atypical distribution (ie: Cauchy,Levy etc) where the moments do not exist/diverge and obtain reliable out of sample predictions , and this is why the test error diverges.
>
> The full derivation exists at reference [9] with an updated version on Arxiv. In our response to Reviewer LTrn we have also included the proof for the correlation traps and are happy to include this in the appendix.
>
>
>
>
> ## QN: Perturbation studies and other metrics like the $L^2$ norm
> As seen by ref [2] in our paper (Progress measures for grokking on real world tasks) $L^2$ has been charecterized as not effectively explaining grokking or generalization , this is also seen in our following experiment, here we perturb by multiplying all the weights,bias parameters by a constant (4) similar to Liu et al (Omnigrok: Grokking Beyond Algorithmic Data) and we see that this threshold does not work (125)
>
>
> | Phase | Train Accuracy (mean ± std) | Test Accuracy (mean ± std) |
> | :------- | --------------------------: | -------------------------: |
> | Pre‑grok | 0.7298 ± 0.0408 | 0.4926 ± 0.0187 |
> | Grok | 1.0000 ± 0.0000 | 0.8879 ± 0.0051 |
> | Antigrok | 1.0000 ± 0.0000 | 0.6184 ± 0.1086 |
>
>
> | Phase    | WeightNorm (mean ± std) |
> | :------- | --------------------------: |
> | Pre‑grok |                17.52 ± 0.00 |
> | Grok     |                16.80 ± 0.22 |
> | Antigrok |                27.64 ± 8.39 |
>
>
>
> | Layer | Pregrok (10²–5×10⁴) mean | Grok (5×10⁴–5×10⁵) mean | Antigrok (>5×10⁵) mean |
> | :----------: | -----------------------: | ----------------------: | ---------------------: |
> | Layer 1 | 8.88 | 2.59 | 1.23 |
> | Layer 2 | 3.84 | 2.77 | 1.49 |
> | **Mean±Std** | **6.36 ± 2.52** | **2.68 ± 0.09** | **1.36 ± 0.13** |
>
>
> #### correlation trap analysis
> | Layer          | Pre‑grok α      | Grok α          | Anti‑grok α     |
> | -------------- | --------------- | --------------- | --------------- |
> | 1              | 0.00            | 0.00            | 7.00            |
> | 3              | 0.00            | 0.00            | 1.17            |
> | 5              | 0.00            | 0.00            | 5.50            |
> | **Mean ± Std** | **0.00 ± 0.00** | **0.00 ± 0.00** | **4.56 ± 3.03** |
>
>
>
> an intuitive way to think of this is because there is essentially a hypersphere of parameter vectors $\mathbf{W} $ which have the same $L^2$ norm, clearly not on points on this hypersphere are generalizing (have high test accuracy).
>
>
> ## QN: Does this apply to other architectures and Regularization methods?
>
> As seen in the paper ref [10] ('Predicting trends in the quality of state-of-the-art neural networks without access to training or testing data' nature paper) This theory has been applied to numerous models and is widely applicable, the scope of our paper is to show that HTSR can effectively analyze grokking and to discover the novel Antigrokking phase.
>
> As seen below we also have experiments run on Modular Addition
>
> ## 1 Modular‑addition transformer Experiment
>
>
> Modular addition (predicting $x+y\bmod P$) was chosen because it was the original algorithmic setting in which grokking occurs. Running for $10^{7}$ steps on a single Quadro P2000 (5 GB) unfortunately went on to roughly one month longer than initially budgeted due to the computational cost...here are the results (which we are happy to add to the appendix and or supplementary material)
>
>
>
> Prime chosen = 113
> ###  Model architecture
>
> | Hyper‑parameter                    | Value                            |
> | ---------------------------------- | -------------------------------- |
> | Layers                             | **1**                            |
> | Model dimension $d_{\text{model}}$ | **128**                          |
> | MLP hidden size $d_{\text{mlp}}$   | **512**                          |
> | Heads × head dim                   | **4 × 32**                       |
> | Context length $n_{\text{ctx}}$    | **3**                            |
> | Activation                         | **ReLU**                         |
> | LayerNorm                          | **disabled** *(use\_ln =False)*  |
> | Vocabulary size                    | **114** (= equals\_token + 1)    |
>
>
>
> ### HTSR results (α)
>
> | Layer                   | Pre‑grok α      | Grok α          | Anti‑grok α     |
> | ----------------------- | --------------- | --------------- | --------------- |
> | embed.embed             | 2.81            | **1.92**        | 3.57            |
> | blocks.0.attn.W\_Q      | 4.04            | **2.86**        | 3.17            |
> | blocks.0.attn.W\_K      | 3.99            | **2.00**        | 3.20            |
> | blocks.0.attn.W\_V      | 4.66            | **1.56**        | 3.87            |
> | blocks.0.attn.out\_proj | 3.85            | **1.57**        | 4.04            |
> | blocks.0.mlp.fc1        | 4.99            | **1.71**        | 4.74            |
> | blocks.0.mlp.fc2        | 5.22            | **2.17**        | 5.03            |
> | unembed.unembed         | 3.27            | **2.37**        | 3.53            |
> | **Mean ± Std**          | **4.10 ± 0.83** | **2.02 ± 0.44** | **3.89 ± 0.68** |
>
> ### 1.6 Accuracy trajectories
>
> | Phase     | Train Acc.      | Test Acc.           |
> | --------- | --------------- | ------------------- |
> | Pre‑grok  | 1.0000 ± 0.0000 | 0.4056 ± 0.2678     |
> | Grok      | 1.0000 ± 0.0000 | **0.9725 ± 0.0166** |
> | Anti‑grok | 1.0000 ± 0.0000 | 0.5978 ± 0.1116     |
>
> Correlation trap analysis is also performed and the table is given below
>
> | Layer                   | Pre‑grok α      | Grok α          | Anti‑grok α     |
> | ----------------------- | --------------- | --------------- | --------------- |
> | embed.embed             | 0.00            | 0.00            | 3.48            |
> | blocks.0.attn.W\_Q      | 0.00            | 0.00            | 3.74            |
> | blocks.0.attn.W\_K      | 0.00            | 0.00            | 3.72            |
> | blocks.0.attn.W\_V      | 0.00            | 0.00            | 2.91            |
> | blocks.0.attn.out\_proj | 0.00            | 0.00            | 3.03            |
> | blocks.0.mlp.fc1        | 0.00            | 0.00            | 5.26            |
> | blocks.0.mlp.fc2        | 0.00            | 0.00            | 6.05            |
> | unembed.unembed         | 0.00            | 0.00            | 4.83            |
> | **Mean ± Std**          | **0.00 ± 0.00** | **0.00 ± 0.00** | **4.13 ± 1.13** |
>
>
> As predicted by HTSR, the global mean exponent $\alpha$ gravitates tightly around 2 during the test‑accuracy peak and deviates from 2 thereafter. Also as we mention in the paper we see the increase of the outlier eigen values from the Marchenko Pastur bulk in all layers during the Antigrok phase.
>
>
> ## Selected References
>  J. Baik, G. Ben Arous & S. Péché, *Ann. Probab.* 33 (2005), 1643–1697.
>  M. Gürbüzbalaban, U. Şimşekli & L. Zhu, *Proc.* 38th ICML (2021), 3964–3975.
>  Charles H. Martin and Michael W. Mahoney. Implicit self-regularization in deep neural networks: Evidence from heavy-tailed spectral analysis, 2021. (JMLR)
>  Charles H. Martin and Christopher Hinrichs SETOL: A Semi-Empirical Theory of (Deep) Learning, 2025 (Arxiv 2507.17912)

---

> > ### Comment · Reviewer_XxTJ · 2025-08-05
> >
> > I would like to thank the authors for their reply and for the new experimental results. While these do strengthen the paper, I do not believe these are enough for publication in NeurIPS. This is primarily for three reasons:
> > 1) The phenomenon of "anti-grokking" has been reported at least two years ago by Varma et al (2023). Although the authors rightfully point out that Varma et al showed ungrokking in a smaller dataset, this is quite effectively explained by the "softmax collapse" described in Prieto et al ("Grokking at the Edge of Numerical Stability"). Thus, I do not consider this finding to be novel.
> > 2) I admit I am not familiar with SETOL, but I am still unconvinced by the theoretical foundations of the work. It is unclear to me what the "quality" of a layer is or why its magnitude would be related to generalisation. Perhaps a reviewer with an HTSR- or SETOL-specific background would be better positioned to judge these links, but the fact remains that the paper does not do a sufficiently good job of explaining its own results for the broader community (who cannot be expected to be experts in HTSR or SETOL).
> > 3) With the two points above in mind, the remaining contribution of this work is a valid and rigorous evaluation of $\alpha$, showing that it is a slightly better progress measure than others (although, again, I am missing some direct quantitative evidence for this).
> >
> > For these reasons, I do not believe the paper meets the requirements for publication in NeurIPS, although I strongly encourage the authors to continue pursuing this promising line of work and resubmit and improved version of the paper.

---

> ### Author Response · Authors · 2025-08-06
> **Rebuttal to Reviewer XxTJ's response**
>
> ## Rebuttal to Reviewer XxTJ's response
>
> ---
> Thank you for your comments. We would like to clarify a key distinction regarding our results and prior work on ungrokking and softmax collapse.
>
> ###  Response part 1.
>
>
> 1. **Our phenomenon differs fundamentally from “ungrokking”** (Varma et al., 2023): ungrokking requires *re-training on a dataset of size* $$D' < D_{\text{crit}}$$ and some weight decay and is explained via circuit formations. We continue training on the *same* dataset with $$D > D_{\text{crit}}$$ yet still observe catastrophic forgetting. This is explained on line 72 to 76 in our paper.
> 2.  **Soft-max collapse cannot arise or explain the collapse in our setting with MSE loss.**
>   Our main experiments use **MSE** loss on the logits (see Appendix A), **without softmax** or cross-entropy. Thus the mechanism described by Prieto et al. cannot arise in our setup, as it depends on numerical issues arising from softmax which is not used here.
> 3. Therefore the reviewer’s claim that ungrokking is the same as "Antigrokking" and that Prieto et al.’s soft-max collapse “quite effectively explains” our result will not hold.
>
> We believe the reviewer's interpretation conflates distinct phenomena. The mechanism we describe is theoretically and empirically different from ungrokking and softmax collapse, as discussed above.
>
> ---
>  Distinguishing our “catastrophic forgetting” from Varma et al.’s “ungrokking”
>
> | **Property** | **Our experiment** | **Varma et al. (2023)** |
> |--------------|--------------------|-------------------------|
> | Dataset size during *second* training phase | Same dataset, $$D > D_{\text{crit}}$$ | New Smaller dataset, $$D' < D_{\text{crit}}$$ |
> | Weight decay | Not required | greater than 0 |
> | Loss & failure mode |Catastrophic generalization loss  after generalizing training set | Catastrophic **forgetting**  **when switching to a new dataset** |
>
> Varma et al. explicitly predict ungrokking *only* when $$D' < D_{\text{crit}}$$:
>
>  Verbatim from Varma et al. page 6 > “… ungrokking should **only** be expected once $$D' < D_{\text{crit}}$$ …”
>
> Our results violate this condition, demonstrating something distinct.
>
> Also on page 12 of **Varma et al.** "**Limitations**: *Necessity of weight decay. The biggest limitation of our explanation is that it relies crucially on
> weight decay, but grokking has been observed even when weight decay is not present (Power et al.,
> 2021; Thilak et al., 2022) (though it is slower and often much harder to elicit (Nanda et al., 2023,
> Appendix D.1)). This demonstrates that our explanation is incomplete.*"
>
> Please note that all of our experiments are in the zero weight decay regime and thus have a further point of difference from the ungrokking experiment.
>
>
> We hope this clarification helps resolve the misunderstanding. Since softmax collapse does not apply under MSE loss, and our observed forgetting occurs under conditions distinct from both Prieto et al. and Varma et al., we believe the reviewers concern may not apply to our setting. We respectfully maintain that our findings represent a *novel* contribution and merit consideration for publication.
>
> ###  Response part 2 SETOL,HTSR.
>
> Alpha quantifies a layer’s quality—that is, its approximate contribution to the model’s generalization accuracy. It is derived in \[9] through a statistical-mechanics analysis and has been validated empirically in a range of earlier studies, including the NeurIPS Spotlight paper *“Temperature Balancing, Layer-wise Weight Analysis, and Neural Network Training.”*  Considerable prior work already addresses HTSR; the present paper extends that line by demonstrating the broad applicability of HTSR and SETOL, introducing a new phase of grokking, and encouraging deeper theoretical and experimental exploration of these ideas, without overly redoing prior art.
>
>
> ### References
> * Prieto, L., Barsbey, M., Mediano, P., & Birdal, T.  *Grokking at the Edge of Numerical Stability*.
> * Varma, V., et al. (2023). *Explaining Grokking through Circuit Efficiency*.

---

### Note · Authors · 2025-08-13

Thank you to all reviewers for the thoughtful discussion.
This paper empirically studies Grokking through the published Heavy-Tailed Self-Regularization (HTSR) lens. Our claims are: (i) a new phase—anti-grokking—and (ii) the HTSR layer-quality metric (power-law exponent α) reliably tracks the three phases (pre → grok → anti). Evidence comes from (1) MNIST, (2) a modular-addition transformer (appendix), and (3) an initialization-scale ablation (appendix). We note that “correlation traps,” found via entry-wise shuffling and KS tests vs the MP baseline, show a significant deviation when generalization collapses.
Reviewers vAL9, pQqz, RbVm, and LTrn indicated their questions were addressed to our delight, acknowledging the modular-addition results, the init-scale ablation, clearer axes/captions, and a proof for shuffling-based outlier detection. Reviewer XxTJ first asked about causality/scope/related work (this was addressed), then later claimed our collapse is ungrokking explained by softmax-collapse. On Aug 6 we replied:
1. Softmax collapse can not be the mechanism here: unlike the ungrokking experiments we use MSE with one-hot targets rather than cross-entropy, so there is no softmax layer.
2. Our collapse occurs while continuing on the same dataset even with zero regularization, whereas “ungrokking” (Varma et al) requires switching to a new, smaller dataset smaller than $D_{crit}$.

We also provided a side-by-side table and pointed to HTSR/SETOL’s derivation of α. Sadly no further reply was received. Notably, Reviewer XxTJ also wrote that “a reviewer with an HTSR- or SETOL-specific background would be better positioned to judge these links”;  we respectfully suggest discounting this specific objection relative to the consensus of the other reviewers and more importantly because HTSR is prior art.
We appreciate that, given the volume of submissions, some reviewers may not be familiar with HTSR or its justification using statistical mechanics ( SETOL, ref [9]). We summarized the key predictions of HTSR of relevance in section 3 of our paper and directed interested reviewers to the prior art, notably refs [9] and [10]. The relevant pieces being the class demarcations of alpha and the concept of correlation traps. Though, we are happy though to elaborate further as necessary.

---

### Decision · Program_Chairs · 2025-09-17

**Decision:**

Reject

**Comment:**

This submission investigates grokking through the spectral properties of weight matrices in trained neural networks. The authors empirically show that the tail exponent in the spectral decay tracks the grokking transition and other training phase changes.
While the observations are interesting, the reviewers raised concerns about the limited experimental scope—particularly the lack of evidence for universality across architectures and datasets—as well as the absence of theoretical insight. These issues were only partially addressed in the rebuttal. The meta-reviewer therefore recommends rejection, and encourages the authors to resubmit the revised manuscript to a future venue.